# HIERARCHICALLY CLUSTERED REPRESENTATION LEARNING

## ABSTRACT

The joint optimization of representation learning and clustering in the embedding space has experienced a breakthrough in recent years. In spite of the advance, clustering with representation learning has been limited to flat-level categories, which oftentimes involves cohesive clustering with a focus on instance relations. To overcome the limitations of flat clustering, we introduce *hierarchically clustered* representation learning (HCRL), which simultaneously optimizes representation learning and hierarchical clustering in the embedding space. Specifically, we place a nonparametric Bayesian prior on embeddings to handle dynamic mixture hierarchies under the variational autoencoder framework, and to adopt the generative process of a hierarchical-versioned Gaussian mixture model. Compared with a few prior works focusing on unifying representation learning and hierarchical clustering, HCRL is the first model to consider a generation of deep embeddings from every component of the hierarchy, not just leaf components. This generation process enables more meaningful separations and mergers of clusters via branches in a hierarchy. In addition to obtaining hierarchically clustered embeddings, we can reconstruct data by the various abstraction levels, infer the intrinsic hierarchical structure, and learn the level-proportion features. We conducted evaluations with image and text domains, and our quantitative analyses showed competent likelihoods and the best accuracies compared with the baselines.

## 1 INTRODUCTION

*Clustering* is one of the most traditional and frequently used machine learning tasks. Clustering models are designed to represent intrinsic data structures, such as latent Dirichlet allocation (Blei et al., 2003). The recent development of *representation learning* has contributed to generalizing model feature engineering, which also enhances data representation (Bengio et al., 2013). Therefore, representation learning has been merged into the clustering models, e.g., variational deep embedding (VaDE) (Jiang et al., 2017). Besides merging representation learning and clustering, another critical line of research is structuring the clustering result, e.g., hierarchical clustering. This paper introduces a unified model enabling nonparametric Bayesian hierarchical clustering with neural-network-based representation learning.

Autoencoder (Rumelhart et al., 1985) is a typical neural network for unsupervised representation learning and achieves a non-linear mapping from a high-dimensional input space to a low-dimensional embedding space by minimizing reconstruction errors. To turn the low-dimensional embeddings into random variables, a variational autoencoder (VAE) (Kingma & Welling, 2014) places a Gaussian prior on the embeddings. The autoencoder, whether it is probabilistic or not, has a limitation in reflecting the intrinsic hierarchical structure of data. For instance, VAE assuming a single Gaussian prior needs to be expanded to suggest an elaborate clustering structure.

Due to the limitations of modeling the cluster structure with autoencoders, prior works combine the autoencoder and the clustering algorithm. While some early cases pipeline just two models, e.g., Huang et al. (2014), a typical merging approach is to model an additional loss, such as a clustering loss, in the autoencoders (Xie et al., 2016; Guo et al., 2017; Yang et al., 2017; Nalisnick et al., 2016; Chu & Cai, 2017; Jiang et al., 2017). These suggestions exhibit gains from unifying the encoding and the clustering, yet they remain at the parametric and flat-structured clustering. A more recent development releases the previous constraints by using the nonparametric Bayesian approach.

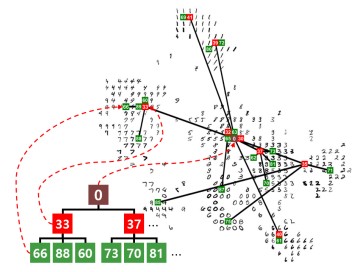

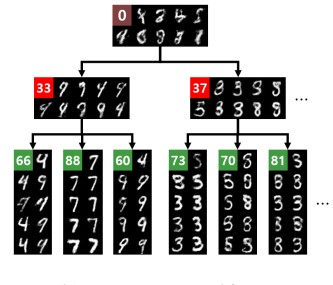

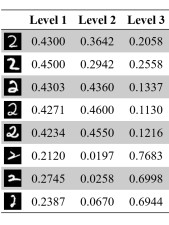

| | Level 1 | Level 2 | Level 3 |
|---|---|---|---|
| 2 | 0.4300 | 0.3642 | 0.2058 |
| 2 | 0.4500 | 0.2942 | 0.2558 |
| 2 | 0.4303 | 0.4360 | 0.1337 |
| 2 | 0.4271 | 0.4600 | 0.1130 |
| 2 | 0.4234 | 0.4550 | 0.1216 |
| 2 | 0.2120 | 0.0197 | 0.7683 |
| 2 | 0.2745 | 0.0258 | 0.6998 |
| 2 | 0.2387 | 0.0670 | 0.6944 |

(a) Hierarchically clustered embeddings     (b) Reconstructed images     (c) Level proportion features of real images

Figure 1: Example of hierarchically clustered embeddings on MNIST with three levels of hierarchy, the reconstructed digits from the hierarchical Gaussian mixture components, and the extracted level proportion features. We marked the mean of a Gaussian mixture component with the colored square, and the digit written inside the square refers to the unique index of the mixture component.

For example, the infinite mixture of VAEs (IMVAE) (Abbasnejad et al., 2017) explores the infinite space for VAE mixtures by looking for an adequate embedding space through sampling, such as the Chinese restaurant process (CRP). Whereas IMVAE remains at the flat-structured clustering, VAE-nested CRP (VAE-nCRP) (Goyal et al., 2017) captures a more complex structure, i.e., a hierarchical structure of the data, by adopting the nested Chinese restaurant process (nCRP) prior (Griffiths et al., 2004) into the cluster assignment of the Gaussian mixture model.

This paper proposes hierarchically clustered representation learning (HCRL) that is a joint model of 1) nonparametric Bayesian hierarchical clustering, and 2) representation learning with neural networks. HCRL extends a previous work on merging flat clustering and representation learning, i.e., VaDE, by incorporating inter-cluster relation modelings. Unlike a previous work of VAE-nCRP, HCRL learns the full spectrum of hierarchical clusterings, such as the level assignment and the level proportion of generating a component hierarchy. These level assignments and proportions were not modeled in VAE-nCRP, so each data instance cannot be analyzed from the perspective of generalization and specialization in a hierarchy. On the contrary, by adding level assignment and proportion modeling, a data instance can be generated from an internal component of the hierarchy, which is limited to the leaf component in VAE-nCRP. Hierarchical mixture density estimation (Vasconcelos & Lippman, 1999), where all internal and leaf components are directly modeled to generate data, is a flexible framework for hierarchical mixture modeling, such as hierarchical topic modeling (Mimno et al., 2007; Griffiths et al., 2004), with regard to the learning of the internal components.

Specifically, HCRL jointly optimizes soft-divisive hierarchical clustering in an embedding space from VAE via two mechanisms. First, HCRL includes a hierarchical-versioned Gaussian mixture model (HGMM) with a mixture of hierarchically organized Gaussian distributions. Then, HCRL sets the prior of embeddings by adopting the generative processes of HGMM. Second, to handle a dynamic hierarchy structure dealing with the clusters of unequal sizes, we explore the infinite hierarchy space by exploiting an nCRP prior. These mechanisms are fused as a unified objective function; this is done rather than concatenating the two distinct models of clustering and autoencoding. The quantitative evaluations focus on density estimation quality and hierarchical clustering accuracy, which shows that HCRL has competent likelihoods and the best accuracies compared with the baselines. When we observe our results qualitatively, we visualize 1) the hierarchical clusterings, 2) the embeddings under the hierarchy modeling, and 3) the reconstructed images from each Gaussian mixture component, as shown in Figure 1. These experiments were conducted by crossing the data domains of texts and images, so our benchmark datasets include MNIST, CIFAR-100, RCV1_v2, and 20Newsgroups.

## 2   Preliminaries

### 2.1   Variational Deep Embedding

Figure 2 presents a graphical representation and a neural architecture of VaDE (Jiang et al., 2017). The model parameters of $\kappa$, $\mu_{1:K}$, and $\sigma^2_{1:K}$, which are a proportion, means, and covariances of

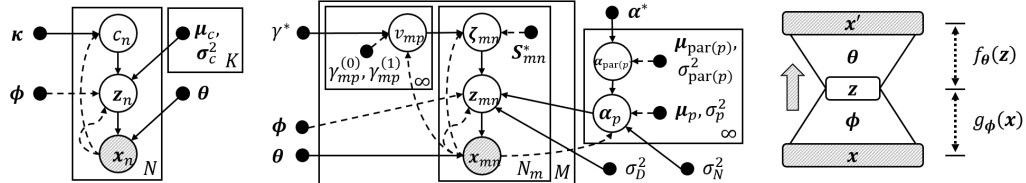

Figure 2: Graphical representation of VaDE (Jiang et al., 2017) (left), VAE-nCRP (Goyal et al., 2017) (center), and neural architecture of both models (right). In the graphical representation, the white/shaded circles represent latent/observed variables. The black dots indicate hyper or variational parameters. The solid lines represent a generative model, and dashed lines represent a variational approximation. A rectangle box means a repetition for the number of times denoted by the bottom right of the box.

mixture components, respectively, are declared outside of the neural network[1]. VaDE trains model parameters to maximize the lower bound of marginal log likelihoods via the mean-field variational inference (Jordan et al., 1999). VaDE uses the Gaussian mixture model (GMM) as the prior, whereas VAE assumes a single standard Gaussian distribution on embeddings. Following the generative process of GMM, VaDE assumes that 1) the embedding draws a cluster assignment, and 2) the embedding is generated from the selected Gaussian mixture component.

VaDE uses an amortized inference as VAE, with a generative and inference networks; $\mathcal{L}(\boldsymbol{x})$ in Equation 1 denotes the evidence lower bound (ELBO), which is the lower bound on the log likelihood. It should be noted that VaDE merges the ELBO of VAE with the likelihood of GMM.

$$\log p(\boldsymbol{x}) \geq \mathcal{L}(\boldsymbol{x}) = \mathbb{E}_q \left[ \log \frac{p(\boldsymbol{c}, \boldsymbol{z}, \boldsymbol{x})}{q(\boldsymbol{c}, \boldsymbol{z}|\boldsymbol{x})} \right] = \mathbb{E}_q \left[ \log \prod_{c=1}^{K} \frac{\kappa_c \mathcal{N}(\boldsymbol{z}|\boldsymbol{\mu}_c, \boldsymbol{\sigma}_c^2 \boldsymbol{I}_J)}{p(c|\boldsymbol{z}) \mathcal{N}(\boldsymbol{z}|\widetilde{\boldsymbol{\mu}}, \widetilde{\boldsymbol{\sigma}}^2 \boldsymbol{I}_J)} + \log p(\boldsymbol{x}|\boldsymbol{z}) \right] \quad (1)$$

## 2.2 Variational Autoencoder nested Chinese Restaurant Process

VAE-nCRP uses the nonparametric Bayesian prior for learning tree-based hierarchies, the nCRP (Griffiths et al., 2004), so the representation could be hierarchically organized. The nCRP prior defines the distributions over children components for each parent component, recursively in a top-down way. The variational inference of the nCRP can be formalized by the nested stick-breaking construction (Wang & Blei, 2009), which is also kept in the VAE setting. The distribution over paths on the hierarchy is defined as being proportional to the product of weights corresponding to the nodes lying in each path. The weight, $\pi_i$, for the $i$-th node follows the Griffiths-Engen-McCloskey (GEM) distribution (Pitman et al., 2002), where $\pi_i$ is constructed as $\pi_i = v_i \prod_{j=1}^{i-1}(1 - v_j), v_i \sim \text{Beta}(1, \gamma)$ by a stick-breaking process. Since the nCRP provides the ELBO with the nested stick-breaking process, VAE-nCRP has a unified ELBO of VAE and the nCRP in Equation 2.

$$\mathcal{L}(\boldsymbol{x}) = \mathbb{E}_q \left[ \log \frac{p(\boldsymbol{v})}{q(\boldsymbol{v}|\boldsymbol{x})} + \log \underbrace{\frac{p(\boldsymbol{\alpha}_{\text{par}(p)}|\boldsymbol{\alpha}^*)p(\boldsymbol{\alpha}_p|\boldsymbol{\alpha}_{\text{par}(p)}, \sigma_N^2)}{q(\boldsymbol{\alpha}_p, \boldsymbol{\alpha}_{\text{par}(p)}|\boldsymbol{x})}}_{(3.1)} \frac{p(\boldsymbol{\zeta}|\boldsymbol{v})}{q(\boldsymbol{\zeta}|\boldsymbol{x})} \underbrace{\frac{p(\boldsymbol{z}|\boldsymbol{\alpha}_p, \boldsymbol{\zeta}, \sigma_D^2)}{q(\boldsymbol{z}|\boldsymbol{x})}}_{(3.2)} + \log p(\boldsymbol{x}|\boldsymbol{z}) \right] \quad (2)$$

Given the ELBO of VAE-nCRP, we recognized a number of potential improvements. First, term (3.1) is for modeling the hierarchical relationship among clusters, i.e., each child is generated from its parent. VAE-nCRP trade-off is the direct dependency modeling among clusters against the mean-field variational approach. This modeling may reveal that the higher clusters in the hierarchy are more difficult to train. Second, in term (3.2), leaf mixture components generate embeddings, which implies that only leaf clusters have direct summarization ability for sub-populations. Additionally, in term (3.2), variance parameter $\sigma_D^2$ is modeled as the hyperparameter shared by all clusters. In other words, only with $J$-dimensional parameters, $\boldsymbol{\alpha}$, for the leaf mixture components, the local density modeling without variance parameters has a critical disadvantage.

For all of these weaknesses, we were able to compensate with the level proportion modeling and HGMM prior. The level assignment generated from the level proportion allows a data instance to

---

[1]Appendix D enumerates the symbols in this paper.

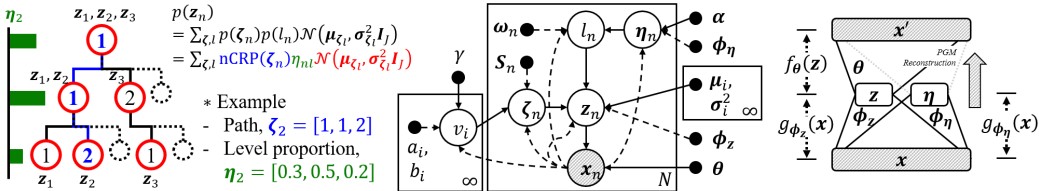

Figure 3: A simple depiction (left) of the key notations, where each numbered circle refers to the corresponding Gaussian mixture component. The graphical representation (center) and the neural architecture (right) of our proposed model, HCRL. The neural architecture of HCRL consists of two probabilistic encoder networks, $g_{\phi_\eta}$ and $g_{\phi_z}$, and one probabilistic decoder network, $f_\theta$.

select among all mixture components. We do not need direct dependency modeling between the parents and their children because all internal mixture components also generate embeddings.

# 3 METHODOLOGY

## 3.1 GENERATIVE PROCESS

The generative process of HCRL resembles the generative process of hierarchical clusterings, such as the hierarchical latent Dirichlet allocation (Griffiths et al., 2004). In detail, the generative process departs from selecting a path $\zeta$, from the nCRP prior (phase 1). Then, we sample a level proportion (phase 2) and a level, $l$ (phase 3), from the sampled level proportion to find the mixture component in the path, and this component of $\zeta_l$ provides the Gaussian distribution for the latent representation (phase 4). Finally, the latent representation is exploited to generate an observed datapoint (phase 5). The below formulas are the generative process with its density functions. In addition, Figure 3 illustrates a graphical representation corresponding to the described generative process. The generative process also presents our formalization of corresponding prior distributions, denoted as $p(\cdot)$, and variational distributions, denoted as $q(\cdot)$, by generation phases. The variational distributions are used in our inference methods called mean-field variational inference (MFVI) (Jordan et al., 1999) as detailed in Section 3.3.

1. Choose a path $\zeta \sim \text{nCRP}(\zeta|\gamma)$
   - $p(\zeta) = \prod_{l=1}^L \pi_{1,\zeta_2,\ldots,\zeta_l}$ where $\pi_{1,\zeta_2,\ldots,\zeta_l} = \prod_{l'=1}^l \{v_{1,\zeta_2,\ldots,\zeta_{l'}}(\prod_{j=1}^{\zeta_{l'}-1}(1-v_{1,\zeta_2,\ldots,j}))\}$,
     $q(\zeta|x) \propto S_{\overline{\zeta}} \triangleq \sum_{\zeta \in \text{child}(\overline{\zeta})} S_\zeta$
2. Choose a level proportion $\eta \sim \text{Dirichlet}(\eta|\alpha)$
   - $p(\eta) = \text{Dir}(\eta|\alpha), q_{\phi_\eta}(\eta|x) = \text{Dirichlet}(\eta|\widetilde{\alpha}) \approx \text{LogisticNormal}(\eta|\widetilde{\mu}_\eta, \widetilde{\sigma}_\eta^2 I_L)$
     where $[\widetilde{\mu}_\eta; \log \widetilde{\sigma}_\eta^2] = g_{\phi_\eta}(x), \widetilde{\alpha}_l = \frac{1}{\widetilde{\sigma}_{\eta_l}^2}(1 - \frac{2}{L} + \frac{e^{-\widetilde{\mu}_{\eta_l}}}{L^2}\sum_{l'} e^{-\widetilde{\mu}_{\eta_{l'}}})$
3. Choose a level $l \sim \text{Multinomial}(l|\eta)$
   - $p(l) = \text{Multinomial}(\eta), q(l|x) = \text{Multinomial}(l|\omega)$
     where $\omega_l \propto \exp\left\{\psi(\widetilde{\alpha}_l) - \psi(\widetilde{\alpha}_0) + \sum_\zeta S_\zeta \left(\sum_{j=1}^J -\frac{1}{2}\log(2\pi\sigma_{\zeta_l,j}^2) - \frac{\widetilde{\sigma}_{z_j}^2}{2\sigma_{\zeta_l,j}^2} - \frac{(\widetilde{\mu}_{z_j}-\mu_{\zeta_l,j})^2}{2\sigma_{\zeta_l,j}^2}\right)\right\}$
4. Choose a latent representation $z \sim \mathcal{N}(z|\mu_{\zeta_l}, \sigma_{\zeta_l}^2 I_J)$
   - $p(z) = \sum_{\zeta,l} p(\zeta|\gamma) \cdot \eta_l \cdot \mathcal{N}(z|\mu_{\zeta_l}, \sigma_{\zeta_l}^2 I_J)$,
     $q_{\phi_z}(z|x) = \mathcal{N}(z|\widetilde{\mu}_z, \widetilde{\sigma}_z^2 I_J)$ where $[\widetilde{\mu}_z; \log \widetilde{\sigma}_z^2] = g_{\phi_z}(x)$
5. Choose an observed datapoint $x \sim \mathcal{N}\left(x|\mu_x, \sigma_x^2 I_D\right)$ where $[\mu_x; \log \sigma_x^2] = f_\theta(z)^2$

## 3.2 NEURAL ARCHITECTURE

The neural architecture of HCRL consists of two probabilistic encoders on $z$ and $\eta$, and one probabilistic decoder on $z$ as shown in the right part of Figure 3. This unbalanced architecture originates

---

[2] We introduce the sample distribution for the real-valued data instances, and Appendix F provides the binary case as well, which we use for MNIST.

from our modeling assumption of $p(\boldsymbol{x}|\boldsymbol{z})$, not $p(\boldsymbol{x}|\boldsymbol{z}, \boldsymbol{\eta})$. The reconstruction design of $\boldsymbol{x}$ depending on the two stochastic variables of $\boldsymbol{z}$ and $\boldsymbol{\eta}$ may lead to a large variance of the reconstruction on $\boldsymbol{x}$. Additionally, we cannot guarantee that both $\boldsymbol{z}$ and $\boldsymbol{\eta}$ contribute to the the reconstruction on $\boldsymbol{x}$ (Chen et al., 2016). Although the decoding structure of $\boldsymbol{\eta}$ is not included explicitly in the neural network architecture of HCRL, we provide the formalization of $p(\boldsymbol{\eta}|\boldsymbol{z})$ in Table 1 according to our generative assumptions. We call this reconstruction process, which is inherently a generative process of the traditional probabilistic graphical model (PGM), *PGM reconstruction* (see the decoding neural network part of Figure 3).

Table 1: Encoding and decoding structure on $\boldsymbol{z}$ and $\boldsymbol{\eta}$ in HCRL. $(s)$ indicates the $s$-th sample.

|  | **Encoding** | **Decoding** |
|---|---|---|
| $\boldsymbol{z}$ | $\boldsymbol{z} \sim q_{\phi_{\boldsymbol{z}}}(\boldsymbol{z}|\boldsymbol{x}), \boldsymbol{z}^{(s)} = g_{\phi_{\boldsymbol{z}}}(\boldsymbol{\epsilon}^{(s)}, \boldsymbol{x})$ | $\boldsymbol{x} \sim p_{\boldsymbol{\theta}}(\boldsymbol{x}|\boldsymbol{z}), \boldsymbol{x}^{(s)} = f_{\boldsymbol{\theta}}(\boldsymbol{z}^{(s)})$ |
| $\boldsymbol{\eta}$ | $\boldsymbol{\eta} \sim q_{\phi_{\boldsymbol{\eta}}}(\boldsymbol{\eta}|\boldsymbol{x}), \boldsymbol{\eta}^{(s)} = g_{\phi_{\boldsymbol{\eta}}}(\boldsymbol{\epsilon}^{(s)}, \boldsymbol{x})$ | $p(\boldsymbol{x}|\boldsymbol{\eta}) \propto \int_{\boldsymbol{v},\boldsymbol{z}} \sum_{\boldsymbol{\zeta},\boldsymbol{l}} p(\boldsymbol{x}|\boldsymbol{z})p(\boldsymbol{z}|\boldsymbol{\zeta},\boldsymbol{l})p(\boldsymbol{l}|\boldsymbol{\eta})p(\boldsymbol{\zeta}|\boldsymbol{v})p(\boldsymbol{v})$ |

## 3.3 Mean-Field Variational Inference

The formal specification can be a factorized probabilistic model as Equation 3, where $\boldsymbol{\Phi} = \{\boldsymbol{v}, \boldsymbol{\zeta}, \boldsymbol{\eta}, \boldsymbol{l}, \boldsymbol{z}\}$ denotes the set of latent variables, and $\mathcal{M}_T$ denotes the set of all nodes in tree $T$.

$$p(\boldsymbol{\Phi}, \boldsymbol{x}) = \prod_{j \notin \mathcal{M}_T} p(v_j|\gamma) \prod_{i \in \mathcal{M}_T} p(v_i|\gamma) \prod_{n=1}^{N} p(\boldsymbol{\zeta}_n|\boldsymbol{v})p(\boldsymbol{\eta}_n|\boldsymbol{\alpha})p(l_n|\boldsymbol{\eta}_n)p(\boldsymbol{z}_n|\boldsymbol{\zeta}_n, l_n)p_{\boldsymbol{\theta}}(\boldsymbol{x}_n|\boldsymbol{z}_n) \quad (3)$$

The proportion and assignment on the mixture components for the $n$-th data instance are modeled by $\boldsymbol{\zeta}_n$ as a path assignment; $\boldsymbol{\eta}_n$ as a level proportion; and $l_n$ as a level assignment. $v$ is a Beta draw used in the stick-breaking construction. The latent variables are inferred through MFVI, and therefore we assume the variational distributions are as Equation 4:

$$q(\boldsymbol{\Phi}|\boldsymbol{x}) = \prod_{j \notin \mathcal{M}_T} p(v_j|\gamma) \prod_{i \in \mathcal{M}_T} q(v_i|a_i, b_i) \prod_{n=1}^{N} q(\boldsymbol{\zeta}_n|\boldsymbol{x}_n)q_{\phi_{\boldsymbol{\eta}}}(\boldsymbol{\eta}_n|\boldsymbol{x}_n)q(l_n|\boldsymbol{\omega}_n, \boldsymbol{x}_n)q_{\phi_{\boldsymbol{z}}}(\boldsymbol{z}_n|\boldsymbol{x}_n) \quad (4)$$

where $q_{\phi_{\boldsymbol{\eta}}}(\boldsymbol{\eta}_n|\boldsymbol{x}_n)$ and $q_{\phi_{\boldsymbol{z}}}(\boldsymbol{z}_n|\boldsymbol{x}_n)$ should be noted because these two variational distributions follow the amortized inference of VAE. $q(\boldsymbol{\zeta}|\boldsymbol{x}) \propto S_{\overline{\boldsymbol{\zeta}}} \triangleq \sum_{\boldsymbol{\zeta} \in \text{child}(\overline{\boldsymbol{\zeta}})} S_{\boldsymbol{\zeta}}$ is the variational distribution over path $\boldsymbol{\zeta}$, where $\text{child}(\overline{\boldsymbol{\zeta}})$ means the set of all full paths that are not in $T$ but include $\overline{\boldsymbol{\zeta}}$ as a sub path. Because we specified both generative and variational distributions, we define the ELBO of HCRL, $\mathcal{L} = \mathbb{E}_q \left[ \log \frac{p(\boldsymbol{\Phi}, \boldsymbol{x})}{q(\boldsymbol{\Phi}|\boldsymbol{x})} \right]$, in Equation 5. Appendix F enumerates the full derivation in detail. We report that the Laplace approximation with the logistic normal distribution is applied to model the prior, $\boldsymbol{\alpha}$, of the level proportion, $\boldsymbol{\eta}$. We choose a conjugate prior of a multinomial, so $p(\boldsymbol{\eta}_n|\boldsymbol{\alpha})$ follows the Dirichlet distribution. To configure the inference network on the Dirichlet prior, the Laplace approximation is used (MacKay, 1998; Srivastava & Sutton, 2017; Hennig et al., 2012).

$$\mathcal{L}(\boldsymbol{x}) = \mathbb{E}_q \left[ \log \frac{p(\boldsymbol{v})}{q(\boldsymbol{v}|\boldsymbol{x})} + \log \frac{p(\boldsymbol{\eta})}{q(\boldsymbol{\eta}|\boldsymbol{x})} + \log \prod_{\boldsymbol{\zeta},l} \frac{p(\boldsymbol{\zeta}|\boldsymbol{v})}{q(\boldsymbol{\zeta}|\boldsymbol{x})} \frac{p(l|\boldsymbol{\eta})}{q(l|\boldsymbol{x})} \frac{p(\boldsymbol{z}|\boldsymbol{\mu}_{\boldsymbol{\zeta}_l}, \boldsymbol{\sigma}_{\boldsymbol{\zeta}_l}^2)}{q(\boldsymbol{z}|\boldsymbol{x})} + \log p(\boldsymbol{x}|\boldsymbol{z}) \right] \quad (5)$$

## 3.4 Training Algorithm of Clustering Hierarchy

This model is formalized according to the stick-breaking process scheme. Unlike the CRP, the stick-breaking process does not represent the direct sampling of the mixture component at the data instance level. Therefore, it is necessary to devise a heuristic algorithm for operations, such as *GROW*, *PRUNE*, and *MERGE*, to refine the hierarchy structure. Appendix C provides details about each operation, together with the overall training algorithm of HCRL. In the below description, an *inner* path and a *full* path refer to the path ending with an internal node and a leaf node, respectively.

- **GROW** expands the hierarchy by creating a new branch under the heavily weighted internal node. Compared with the work of Wang & Blei (2009), we modified GROW to first sample a path, $\overline{\boldsymbol{\zeta}}^*$, proportional to $\sum_n q(\boldsymbol{\zeta}_n = \overline{\boldsymbol{\zeta}}^*)$, and then to grow the path if the sampled path is an inner path.

- **PRUNE** cuts a randomly sampled minor full path, $\overline{\boldsymbol{\zeta}}^*$, satisfying $\frac{\sum_n q(\boldsymbol{\zeta}_n = \overline{\boldsymbol{\zeta}}^*)}{\sum_{n,\overline{\boldsymbol{\zeta}}} q(\boldsymbol{\zeta}_n = \overline{\boldsymbol{\zeta}})} < \delta$, where $\delta$ is the pre-defined threshold. If the removed leaf node of the full path is the last child of the parent node, we also recursively remove the parent node.

- **MERGE** combines two full paths, $\overline{\boldsymbol{\zeta}}^{(i)}$ and $\overline{\boldsymbol{\zeta}}^{(j)}$, with similar posterior probabilities, measured by $J(\overline{\boldsymbol{\zeta}}^{(i)}, \overline{\boldsymbol{\zeta}}^{(j)}) = \boldsymbol{q}_i \boldsymbol{q}_j^T / |\boldsymbol{q}_i||\boldsymbol{q}_j|$, where $\boldsymbol{q}_i = [q(\boldsymbol{\zeta}_1 = \overline{\boldsymbol{\zeta}}^{(i)}), \cdots, q(\boldsymbol{\zeta}_N = \overline{\boldsymbol{\zeta}}^{(i)})]$.

## 4 EXPERIMENTS

### 4.1 DATASETS AND BASELINES

**Datasets:** We used various hierarchically organized benchmark datasets as well as MNIST.

- **MNIST (LeCun et al., 1998):** 28x28x1 handwritten image data, with 60,000 train images and 10,000 test images. We reshaped the data to 784-d in one dimension.
- **CIFAR-100 (Krizhevsky & Hinton, 2009):** 32x32x3 colored images with 20 coarse and 100 fine classes. We used 3,072-d flattened data with 50,000 training and 10,000 testing.
- **RCV1_v2 (Lewis et al., 2004):** The preprocessed text of the Reuters Corpus Volume. We preprocessed the text by selecting the top 2,000 tf-idf words. We used the hierarchical labels up to the 4-level, and the multi-labeled documents were removed. The final preprocessed corpus consists of 11,370 training and 10,000 testing documents randomly sampled from the original test corpus.
- **20Newsgroups (Lang, 1995):** The benchmark text data extracted from 20 newsgroups, consisting 11,314 training and 7,532 testing documents. We also labeled by 4-level following the annotated hierarchical structure. We preprocessed the data through the same process as that of RCV1_v2.

**Baselines:** We completed our evaluation in two aspects: 1) optimizing the density estimation, and 2) clustering the hierarchical categories. First, we evaluated HCRL from the density estimation perspective by comparing it with diverse flat clustered representation learning models, and VAE-nCRP. Second, we tested HCRL from the accuracy perspective by comparing it with multiple divisive hierarchical clusterings. The below is the list of baselines. We also added the two-stage pipeline approaches, where we trained features from VaDE first and then applied the hierarchical clusterings. We reused the open source codes[3] provided by the authors for several baselines, such as IDEC, DCN, VAE-nCRP, and SSC-OMP.

1. **Variational Autoencoder (VAE) (Kingma & Welling, 2014)**
2. **Variational Deep Embedding (VaDE) (Jiang et al., 2017)**
3. **Improved Deep Embedded Clustering (IDEC) (Guo et al., 2017):** improves DEC (Xie et al., 2016) by attatching decoder structure. We use the code by the authors.
4. **Deep Clustering Network (DCN) (Yang et al., 2017):** optimizes the K-means-related cost defined on the embedding space. We used the open source code provided by the authors.
5. **Infinite Mixture of Variational Autoencoders (IMVAE) (Abbasnejad et al., 2017):** searches for the infinite embedding space by using a Bayesian nonparametric prior.
6. **Variational Autoencoder - nested Chinese Restaurant Process (VAE-nCRP) (Goyal et al., 2017):** We used the open source code provided by the authors.
7. **Hierarchical K-means (HKM) (Nister & Stewenius, 2006):** performs K-means (Lloyd, 1982) recursive in a top-down way.
8. **Mixture of Hierarchical Gaussians (MOHG) (Vasconcelos & Lippman, 1999):** infers the level-specific mixture of Gaussians.
9. **Recursive Gaussian Mixture Model (RGMM):** runs GMM recursively in a top-down manner.
10. **Recursive Scalable Sparse Subspace Clustering by Orthogonal Matching Pursuit (RSS-COMP):** performs SSC-OMP (You et al., 2016) recursively for hierarchical clustering. SSC-OMP is a well-known methods for image clustering, and we used the open source code.

### 4.2 QUANTITATIVE ANALYSIS

We used two measures to evaluate the learned representations in terms of the density estimations: 1) negative log likelihood (NLL), and 2) reconstruction errors (REs). Autoencoder models, such as

---

[3]https://github.com/XifengGuo/IDEC (IDEC); https://github.com/boyangumn/DCN (DCN); https://github.com/prasoongoyal/bnp-vae (VAE-nCRP); http://vision.jhu.edu/code/ (SSC-OMP)

IDEC and DCN, were tested only for the REs. The NLL is estimated with 100 samples. Table 2 indicates that HCRL is best in the NLL and is competent in the REs which means that the hierarchically clustered embeddings preserve the intrinsic raw data structure.

Table 2: Test set performance of the negative log likelihood (NLL) and the reconstruction errors (REs). Replicated ten times, and the best in bold. $P^{\dagger} < 0.05$ (Student's t-test). *Model-L#* means that the model trained with the #-depth hierarchy.

| Model | MNIST | | CIFAR-100 | | RCV1_v2 | | 20Newsgroups | |
|---|---|---|---|---|---|---|---|---|
| | NLL | REs | NLL | REs | NLL | REs | NLL | REs |
| VAE | 230.71 | 10.46 | 1960.06 | 57.54 | 2559.46 | 1434.59 | 2735.80 | 1788.22 |
| VaDE | 217.20 | 10.35 | 1921.85 | 53.60 | 2558.32 | 1426.38 | 2733.46 | 1782.86 |
| IDEC | N/A | 12.75 | N/A | 64.09 | N/A | 1376.26 | N/A | $1660.61^{\dagger}$ |
| DCN | N/A | 11.30 | N/A | 44.26 | N/A | 1361.98 | N/A | 1691.17 |
| IMVAE | 296.57 | 10.69 | 1992.83 | $40.45^{\dagger}$ | 2566.01 | 1387.02 | 2722.81 | 1718.08 |
| VAE-nCRP-$L3$ | 718.78 | 32.67 | 2969.62 | 198.66 | 2642.88 | 1538.42 | 2712.28 | 1680.56 |
| VAE-nCRP-$L4$ | 721.00 | 32.53 | 2950.73 | 198.97 | 2646.48 | 1542.81 | 2713.58 | 1680.71 |
| HCRL-$L3$ | $\mathbf{203.24}^{\dagger}$ | $\mathbf{8.70}^{\dagger}$ | $1843.40^{\dagger}$ | 50.44 | $2554.50^{\dagger}$ | 1395.05 | 2726.75 | 1828.71 |
| HCRL-$L4$ | $203.91^{\dagger}$ | $\mathbf{8.16}^{\dagger}$ | $\mathbf{1849.13}^{\dagger}$ | 50.47 | $\mathbf{2535.43}^{\dagger}$ | **1353.34** | **2702.88** | 1711.30 |

VaDE generally performed better than VAE did, whereas other flat clustered representation learning models tended to be slightly different for each dataset. HCRL showed overall competent performance and better results with a deeper hierarchy of level four than of level three, which implies that capturing the deeper hierarchical structure is likely to be useful for the density estimation.

Additionally, we evaluated hierarchical clustering accuracies by following Xie et al. (2016), except for MNIST that is flat structured. Table 3 points out that HCRL has significantly better micro-averaged F-scores compared with every baseline. HCRL is able to reproduce the ground truth hierarchical structure of the data, and this trend is consistent when HCRL compared with the pipelined model, such as VaDE with a clustering model. The result of the comparisons with the clustering models, such as HKM, MOHG, RGMM, and RSSCOMP, is interesting because it experimentally proves that the joint optimization of hierarchical clustering in the embedding space improves hierarchical clustering accuracies. HCRL also presented better hierarchical accuracies than VAE-nCRP. We conjecture the reasons for the modeling aspect of VAE-nCRP: 1) the simplified prior modeling on the variance of the mixture component as just constants, and 2) the non-flexible learning of the internal components.

Table 3: Hierarchical clustering accuracies with F-scores, on CIFAR-100 with a depth of three, RCV1_v2 with a depth of four, and 20Newsgroups with a depth of four. Replicated ten times, and a confidence interval with 95%. Best in bold.

| Model | CIFAR-100 | RCV1_v2 | 20Newsgroups |
|---|---|---|---|
| HKM | $0.1620_{\pm 0.0077}$ | $0.2564_{\pm 0.0679}$ | $0.4088_{\pm 0.0426}$ |
| MOHG | $0.0846_{\pm 0.0378}$ | $0.1026_{\pm 0.0135}$ | $0.0402_{\pm 0.0119}$ |
| RGMM | $0.1686_{\pm 0.0115}$ | $0.2743_{\pm 0.0521}$ | $0.4351_{\pm 0.0369}$ |
| RSSCOMP | $0.1461_{\pm 0.0228}$ | $0.2657_{\pm 0.0545}$ | $0.2953_{\pm 0.0474}$ |
| VAE-nCRP | $0.2011_{\pm 0.0076}$ | $0.4128_{\pm 0.0242}$ | $0.5584_{\pm 0.0267}$ |
| VaDE+HKM | $0.1637_{\pm 0.0116}$ | $0.3308_{\pm 0.0664}$ | $0.4850_{\pm 0.0558}$ |
| VaDE+MOHG | $0.1659_{\pm 0.0155}$ | $0.4227_{\pm 0.0927}$ | $0.4915_{\pm 0.0713}$ |
| VaDE+RGMM | $0.1806_{\pm 0.0132}$ | $0.3858_{\pm 0.0615}$ | $0.4095_{\pm 0.0651}$ |
| VaDE+RSSCOMP | $0.1923_{\pm 0.0211}$ | $0.2718_{\pm 0.0444}$ | $0.2905_{\pm 0.0431}$ |
| HCRL | $\mathbf{0.2245}_{\pm 0.0137}$ | $\mathbf{0.4553}_{\pm 0.0295}$ | $\mathbf{0.6008}_{\pm 0.0973}$ |

## 4.3 QUALITATIVE ANALYSIS

**MNIST:** In Figure 1, the digits $\{4, 7, 9\}$ and the digits $\{3, 8\}$ are grouped together with a clear hierarchy, which was consistent between HCRL and VaDE. Also, some digits $\{0, 4, 2\}$ in a round form are grouped, together, in HCRL. In addition, among the reconstructed digits from the hierarchical mixture components, the digits generated from the root have blended shapes from 0 to 9, which is natural considering the root position.

**CIFAR-100:** Figure 4 shows the hierarchical clustering results on CIFAR-100. Given that there were no semantic inputs from the data, the color was dominantly reflected in the clustering criteria. However, if one observes the second hierarchy, the scene images of the same sub-hierarchy are semantically consistent, although the background colors are slightly different.

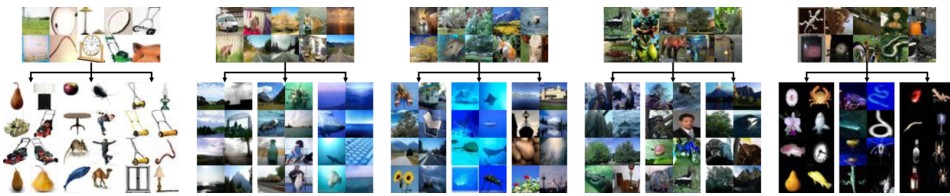

Figure 4: Example extracted sub-hierarchies on CIFAR-100

**RCV1_v2:** Figure 5 shows the embedding of RCV1_v2. VAE and VaDE show no hierarchy, and close sub-hierarchies are distantly embedded. VAE-nCRP guides the internal mixture components to be agglomerated at the center, and the cause of agglomeration is the generative process of VAE-nCRP, where the parameter of the internal components are inferred without direct information from data. HCRL shows a clear separation between the sub-hierarchy without the agglomeration.

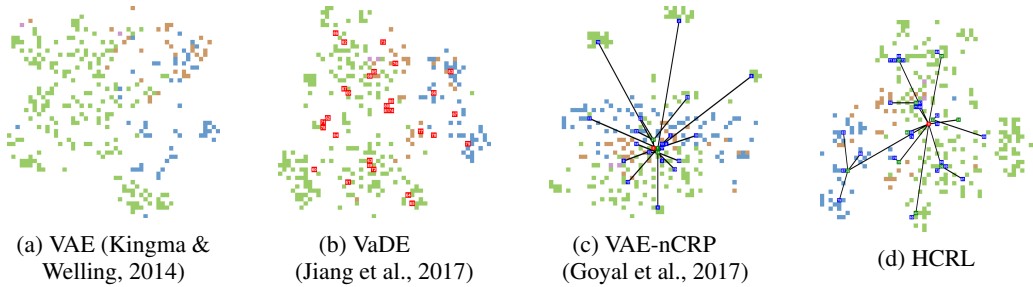

(a) VAE (Kingma &
Welling, 2014)

(b) VaDE
(Jiang et al., 2017)

(c) VAE-nCRP
(Goyal et al., 2017)

(d) HCRL

Figure 5: Comparison of embeddings on RCV1_v2, plotted using t-SNE (Maaten & Hinton, 2008). We mark the mean of a mixture component with a numbered square, colored in {red} for VaDE, {red (root), green (internal), blue (leaf)} for VAE-nCRP and HCRL. The first-level sub-hierarchies are indicated with four colors.

**20Newsgroups:** Figure 6 shows the example sub-hierarchies on 20Newsgroups. We enumerated topic words from documents with top-five likelihoods for each cluster, and we filtered the words by tf-idf values. We observe relatively more general contents in the internal clusters than in the leaf clusters of each internal cluster.

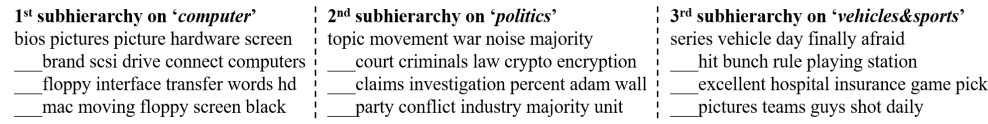

Figure 6: Example extracted sub-hierarchies on 20Newsgroups

## 5 CONCLUSION

In this paper, we have introduced a hierarchically clustered representation learning framework for the hierarchical mixture density estimation on deep embeddings. HCRL aims at encoding the relations among clusters as well as among instances to preserve the internal hierarchical structure of data. The main differentiated features of HCRL are 1) the crucial assumption regarding the internal mixture components for having the ability to generate data directly, and 2) the unbalanced autoencoding neural architecture for the level proportion modeling as the encoding structure, and the probabilistic model as the decoding structure. From the modeling and the evaluation, we found that HCRL enables the improvements due to the high flexibility modeling compared with the baselines.

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

## A    SYNTHETIC DEMO

We created a synthetic dataset that has a hierarchical structure and is sampled from the 50-dimensional Gaussian distributions, presented in Figure 7. The hierarchy, which has a branch factor of two and a depth of four, has a total of eight leaf clusters. Figure 7a shows the raw synthetic dataset in the input space of $\mathbb{R}^{50}$, and after running HCRL, we plot the hierarchically clustered embeddings in the latent space in Figure 7b. In addition to the embeddings, we also present a confidence ellipse with dashed lines for each learned Gaussian mixture component. Because the root component is involved in generating all of the data, it forms a large ellipse, while the leaf component summarizes the local density, so the small ellipse is learned.

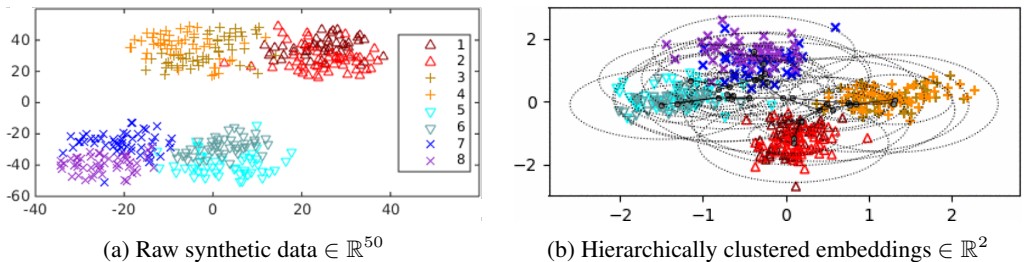

(a) Raw synthetic data $\in \mathbb{R}^{50}$          (b) Hierarchically clustered embeddings $\in \mathbb{R}^2$

Figure 7: Synthetic data in the input space of $\mathbb{R}^{50}$ (left), which is visualized via t-SNE (Maaten & Hinton, 2008), and hierarchically clustered embeddings in the latent space of $\mathbb{R}^2$ (right). We additionally show a 95% confidence ellipse with a dashed line for each Gaussian mixture component.

We show how the above embeddings learned to be hierarchically clustered in the latent space during training in Figure 8. In the learning mechanism of HCRL, we can observe the hierarchically clustered embeddings from a major deviation to a minor deviation in the data over iterations.

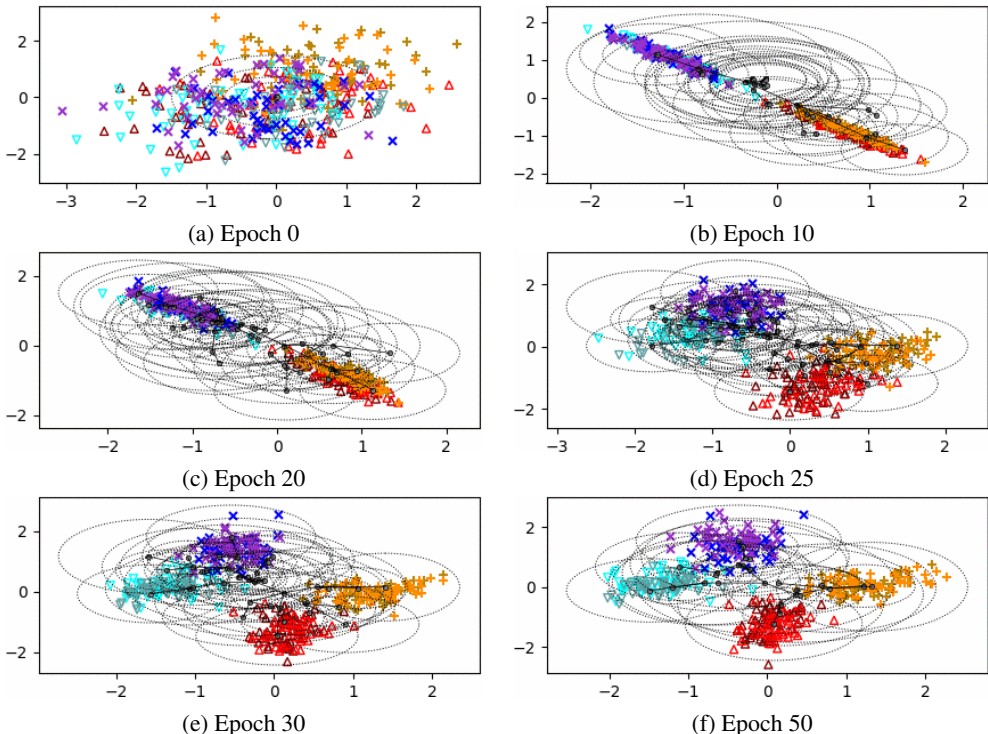

(a) Epoch 0          (b) Epoch 10

(c) Epoch 20          (d) Epoch 25

(e) Epoch 30          (f) Epoch 50

Figure 8: The process by which the embeddings of the synthetic data are learned. The dashed ellipse corresponds to the 95% contour of the learned Gaussian mixture component, whose mean is marked as the gray circle.

## B  Experimental Settings

We conducted experiments for all autoencoder-based models with a neural architecture whose encoder network was set as fully connected layers with dimensions $D$-2000-2000-500-$J$ for $z$, and $D$-10-10-$L$ for $\eta$, and the decoder network is a mirror of the encoder network for $z$. The hyperparameters of HCRL given by users, $\gamma$ and $\alpha$, was set to 1.0, and a vector of all entries 1 sized of $L$, respectively. We used the Adam optimizer (Kingma & Ba, 2014) with an init learning rate of 0.001 for MNIST dataset and 0.0001 for other datasets. Meanwhile, VAE-nCRP is targeted for grouped data. For experiments with our non-grouped datasets, we treated the group instance as a group instance having a single data instance. For parametric hierarchical clustering models, we gave the branch factor as the input parameter, $[1, 20, 5]$, $[1, 4, 7, 9]$, and $[1, 6, 4, 3]$, for CIFAR-100, RCV1_v2, and 20Newsgroups, respectively. For VaDE, we set the number of clusters to the number of leaf clusters; 100 for CIFAR-100, 252 for RCV1_v2, and 72 for 20Newsgroups.

## C  Algorithms

### C.1  Training Algorithm

Algorithm 1 summarizes the overall algorithm for HCRL. The tree-based hierarhcy $T$ is defined as $(\mathbb{N}, \mathbb{P})$, where $\mathbb{N}$ and $\mathbb{P}$ denote a set of nodes and paths, respectively. We refer to the node at level $l$ lying on path $\zeta$, as $N(\zeta_{1:l}) \in \mathbb{N}$. The defined paths, $\mathbb{P}$, consist of full paths (ending at a leaf node), $\mathbb{P}_{\text{full}}$, and inner paths (ending at an internal node), $\mathbb{P}_{\text{inner}}$, as a union set.

Algorithm 1 selects an operation out of three operations: *GROW*, *PRUNE*, and *MERGE*. The GROW algorithm is executed for every specific iteration period, $t_G$. After ellapsing $t_b$ iterations since performing the GROW operation, we begin to check whether the PRUNE or MERGE operation should be performed. We prioritize the PRUNE operation first, and if the condition of performing PRUNE is not satisfied, we check for the MERGE operation next. After performing any operation, we initialize $n_b$ to 0, which is for locking the changed hierarchy during minimum $t_b$ iterations to be fitted to the training data.

---

**Algorithm 1** Training for Hierarchically Clustered Representation Learning

**Input:**  Training examples $x$; the tree-based hierarchy depth, $L$; period of performing GROW, $t_{\text{grow}}$; minimum number of epochs locking the hierarchy, $t_{\text{lock}}$; operation-related thresholds $\delta_{\text{prune}}$, $\delta_{\text{merge}}$; a queue whose element is the set of changed paths, $\mathbb{Q}$; the number of training epochs, $E$; maximum length of $\mathbb{Q}$, $Q_{\text{max}}$; grow scale, $s_{\text{grow}}$

**Output:** $T^{(E)}, \phi_z, \phi_\eta, \theta, \omega, \{a_i, b_i, \mu_i, \sigma_i^2\}_{i \in \mathbb{M}_{T^{(E)}}}$

1:  $\mu_{\overline{\zeta}_{1:L}}, \sigma^2_{\overline{\zeta}_{1:L}} \leftarrow$ Initialize $L$ Gaussian mixture components
2:  $T^{(0)} \leftarrow$ Initialize the tree-based hierarchy having a single path with $\mu_{\overline{\zeta}_{1:L}}, \sigma^2_{\overline{\zeta}_{1:L}}$
3:  $n_{\text{lock}} \leftarrow 0$ // for counting the number of epochs, where the hierarchy has not changed
4:  **for** each epoch $e = 1, \cdots, E$ **do**
5:    $\phi_z, \phi_\eta, \theta \leftarrow$ Update the network weight parameters using gradients $\nabla_{\phi_z, \phi_\eta, \theta} \mathcal{L}(x)$
6:    $\{a_i, b_i, \mu_i, \sigma_i^2\}_{i \in \mathbb{M}_{T^{(e-1)}}} \leftarrow$ Update node-specific params. using gradients $\nabla_{a, b, \mu, \sigma^2} \mathcal{L}(x)$
7:    Update other variational parameters using gradients $\nabla \mathcal{L}(x)$
8:    **if** $\text{mod}(e, t_{\text{grow}}) = 0$ **then**
9:      $T^{(e)}, \mathbb{Q} \leftarrow$ GROW$(T^{(e-1)}, \mathbb{Q}, s_{\text{grow}}, Q_{\text{max}})$ // See Algorithm 2
10:    **end if**
11:    **if** $T^{(e)} = T^{(e-1)}$ and $n_{\text{lock}} \geq t_{\text{lock}}$ **then**
12:      $T^{(e)}, \mathbb{Q} \leftarrow$ PRUNE$(T^{(e-1)}, \mathbb{Q}, \delta_{\text{prune}})$ // See Algorithm 3
13:      **if** $T^{(e)} = T^{(e-1)}$ **then** $T^{(e)}, \mathbb{Q} \leftarrow$ MERGE$(T^{(e-1)}, \mathbb{Q}, \delta_{\text{merge}}, Q_{\text{max}})$ // See Algorithm 4
14:    **end if**
15:    **if** $T^{(e)} \neq T^{(e-1)}$ **then** $n_{\text{lock}} \leftarrow 0$ **else** $n_{\text{lock}} \leftarrow n_{\text{lock}} + 1$
16: **end for**

---

### C.2 Algorithm for Grow Operation

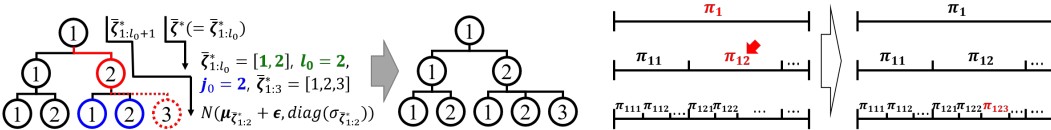

Figure 9: The illustration of GROW operation

The GROW operation expands the hierarchy by creating a new branch under the heavily weighted internal node. Compared with the work from Wang & Blei (2009), we modify GROW to firstly sample a path according to $\sum_{n=1}^{N} q(\zeta_n = \overline{\zeta})$, and then grow the path if the sampled path is an inner path. When we create the new Gaussian mixture component, we initialize the parameters of a corresponding Gaussian distribution depending on the mean and the variance of the parent node, as shown in line 10 of Algorithm 2.

---

**Algorithm 2** GROW Operation

1: **function** GROW($T, \mathbb{Q}, s_{\text{grow}}, Q_{\max}$)
2:      $J(\overline{\zeta}) \leftarrow \sum_{n=1}^{N} q(\zeta_n = \overline{\zeta})$ for $\overline{\zeta} \in \mathbb{P}$ // Calculate the measure
3:      Sample a path $\overline{\zeta}^*$ with probability $\frac{J(\overline{\zeta}^*)}{\sum_{\overline{\zeta}} J(\overline{\zeta})}$
4:      $Q' \leftarrow \phi$ // Temporary set of changed paths in this epoch
5:      **if** $\overline{\zeta}^* \in \mathbb{P}_{\text{inner}}$ and $\overline{\zeta}^* \notin Q$ s.t. $Q \in \mathbb{Q}$ **then**
6:          $l_0 \leftarrow |\overline{\zeta}^*|$
7:          **for** $l' = l_0, \cdots, L-1$ **do**
8:              $j_0 \leftarrow$ Maximum index for the child node whose parent path is $\overline{\zeta}^*_{1:l'}$
9:              $\overline{\zeta}^*_{1:l'+1} \leftarrow \left[\overline{\zeta}^*_{1:l'}, j_0 + 1\right]$
10:            $N(\overline{\zeta}^*_{1:l'+1}) \leftarrow \mathcal{N}(\mu_{\overline{\zeta}^*_{1:l'}} + \epsilon, \text{diag}(\sigma^2_{\overline{\zeta}^*_{1:l'}}))$ where $\epsilon \sim \mathcal{N}(\mathbf{0}, n_g \mathbf{I}_J)$
11:              $Q' \leftarrow Q' \cup \{\overline{\zeta}^*_{1:l'+1}\}$
12:              **if** $l' < L-1$ **then**
13:                  $\mathbb{P}_{\text{inner}} \leftarrow \mathbb{P}_{\text{inner}} \cup \{\overline{\zeta}^*_{1:l'+1}\}$
14:              **else**
15:                  $\mathbb{P}_{\text{full}} \leftarrow \mathbb{P}_{\text{full}} \cup \{\overline{\zeta}^*_{1:L}\}$
16:              **end if**
17:          **end for**
18:      **end if**
19:      *enqueue $Q'$ to $\mathbb{Q}$*
20:      **while** $Q_{\max} < |\mathbb{Q}|$ **do** *dequeue $\mathbb{Q}$*
21:      $\mathbb{P} \leftarrow \mathbb{P}_{\text{full}} \cup \mathbb{P}_{\text{inner}}$
22:      $T \leftarrow (\mathbb{N}, \mathbb{P})$
23:      **return** $T, \mathbb{Q}$
24: **end function**

---

## C.3 Algorithm for Prune Operation

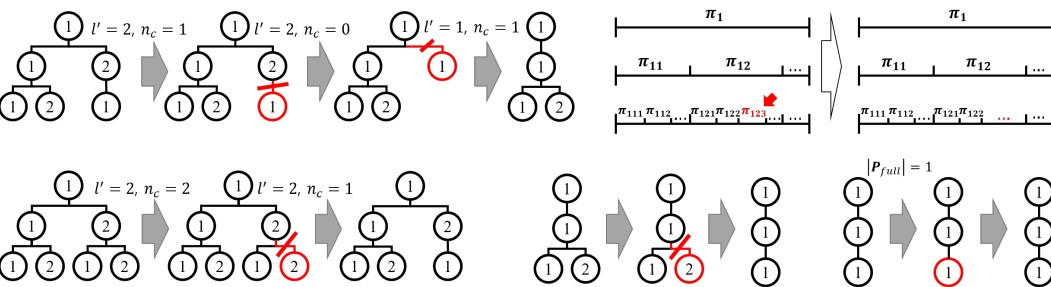

Figure 10: The illustration of PRUNE operation

The PRUNE operation cuts a minor path, which is sampled according to $\sum_{n=1}^{N} q(\zeta_n = \overline{\zeta})$ among the full paths satisfying $\sum_{n=1}^{N} q(\zeta_n = \overline{\zeta}) < \delta$, where $\delta$ is the pre-defined threshold parameter. If the removed leaf node of the full path is the last child of the parent node, we also recursively remove the parent node as shown in the upper case of Figure 10.

---

**Algorithm 3** PRUNE Operation

1: **function** PRUNE($T, \mathbb{Q}, \delta_{\text{prune}}$)
2:     $J(\overline{\zeta}) \leftarrow \sum_{n=1}^{N} q(\zeta_n = \overline{\zeta}), \overline{\zeta} \in \mathbb{P}$ // Calculate the measure
3:     $\Omega \leftarrow \{\overline{\zeta} \mid \overline{\zeta} \in \mathbb{P}_{\text{full}}, \frac{J(\overline{\zeta})}{\sum_{\overline{\zeta}'} J(\overline{\zeta}')} < \delta_{\text{prune}}\}$
4:     Randomly sample a full path $\overline{\zeta}^* \sim \Omega$
5:     **if** $|\mathbb{P}_{\text{full}}| > 1$ and $\overline{\zeta}^* \notin Q$ s.t. $Q \in \mathbb{Q}$ **then**
6:         $\mathbb{P}_{\text{full}} \leftarrow \mathbb{P}_{\text{full}} \backslash \{\overline{\zeta}^*_{1:L}\}$
7:         **for** $l' = L-1, \cdots, 1$ **do**
8:             $\text{N}(\overline{\zeta}^*_{1:l'+1}) \leftarrow \phi$
9:             **if** $l' < L-1$ **then**
10:                $\mathbb{P}_{\text{inner}} \leftarrow \mathbb{P}_{\text{inner}} \backslash \{\overline{\zeta}^*_{1:l'+1}\}$
11:            **end if**
12:            $n_c \leftarrow$ Number of the children nodes whose parent path is $\overline{\zeta}^*_{1:l'}$
13:            **if** $n_c > 0$ **then**
14:                **break**
15:            **end if**
16:        **end for**
17:    **end if**
18:    $\mathbb{P} \leftarrow \mathbb{P}_{\text{full}} \cup \mathbb{P}_{\text{inner}}$
19:    $T \leftarrow (\mathbb{N}, \mathbb{P})$
20:    **return** $T, \mathbb{Q}$
21: **end function**

---

## C.4 Algorithm for Merge Operation

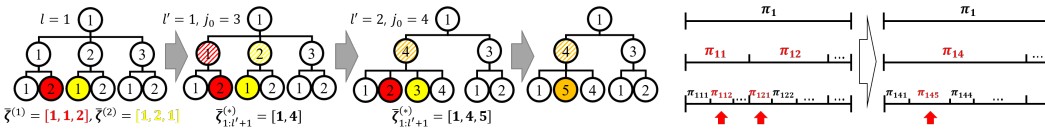

Figure 11: The illustration of MERGE operation

The MERGE operation combines two full paths with similar posterior probabilities, measured by $J(\overline{\zeta}^{(i)}, \overline{\zeta}^{(j)}) = \boldsymbol{q}_i \boldsymbol{q}_j^T / |\boldsymbol{q}_i||\boldsymbol{q}_j|$, where $\boldsymbol{q}_i = [q(\zeta_1 = \overline{\zeta}^{(i)}), ..., q(\zeta_N = \overline{\zeta}^{(i)})]$. We merged two Gaussian components by following Ueda et al. (1999). The specific meaning of combining the two paths is merging the paired two Gaussian distributions lying on the two paths by level, if the two Gaussian distribtions are different. The estimation of merged Gaussian parameters, $\boldsymbol{\mu}$ and $\boldsymbol{\sigma}$, is the weighted summation of two subject Gaussian parameters. The propbability of the node at level $l$ lying on a path $\boldsymbol{\zeta}$ given $\boldsymbol{x}$, $p(\zeta_l|\boldsymbol{x})$, is proportional to $\sum_n \{q(l_n = l) \cdot \sum_{\boldsymbol{\zeta} \in \Lambda} q(\zeta_n = \boldsymbol{\zeta})\}$, where $\Lambda = \{\boldsymbol{\zeta}'|\zeta_l' = \zeta_l \text{ and } \boldsymbol{\zeta}' \in \mathbb{P}_{\text{full}}\}$.

---

**Algorithm 4** MERGE Operation

1: **function** MERGE$(T, \mathbb{Q}, \delta_{\text{merge}}, Q_{\text{max}})$

2:     $J(\overline{\zeta}^{(i)}, \overline{\zeta}^{(j)}) \leftarrow \frac{\boldsymbol{q}_i \boldsymbol{q}_j^T}{|\boldsymbol{q}_i||\boldsymbol{q}_j|}$ s.t. $\boldsymbol{q}_i = [q(\zeta_1 = \overline{\zeta}^{(i)}), ..., q(\zeta_N = \overline{\zeta}^{(i)})]$ // Calculate the measure

3:     $\Omega \leftarrow \{(\overline{\zeta}^{(i)}, \overline{\zeta}^{(j)}) \mid J(\overline{\zeta}^{(i)}, \overline{\zeta}^{(j)}) \geq \delta_{\text{merge}}, \{\overline{\zeta}^{(i)}, \overline{\zeta}^{(j)}\} \subset \mathbb{P}_{\text{full}}\}$

4:     Randomly sample a pair of paths $(\overline{\zeta}^{(1)}, \overline{\zeta}^{(2)}) \sim \Omega$

5:     $Q' \leftarrow \phi$ // Temporary set of changed paths in this epoch

6:     **if** $\{\overline{\zeta}^{(1)}, \overline{\zeta}^{(2)}\} \nsubseteq Q$ s.t. $Q \in \mathbb{Q}$ **then**

7:         $l \leftarrow$ Maximum level of nodes shared by $\overline{\zeta}^{(1)}, \overline{\zeta}^{(2)}$

8:         $\overline{\zeta}^*_{1:l} \leftarrow \overline{\zeta}^{(1)}_{1:l}$

9:         **for** $l' = l, \cdots, L - 1$ **do**

10:             $\boldsymbol{\mu}_{(1)} \leftarrow \boldsymbol{\mu}_{\overline{\zeta}^{(1)}_{1:l'+1}}, \boldsymbol{\sigma}^2_{(1)} \leftarrow \boldsymbol{\sigma}^2_{\overline{\zeta}^{(1)}_{1:l'+1}}, \boldsymbol{\mu}_{(2)} \leftarrow \boldsymbol{\mu}_{\overline{\zeta}^{(2)}_{1:l'+1}}, \boldsymbol{\sigma}^2_{(2)} \leftarrow \boldsymbol{\sigma}^2_{\overline{\zeta}^{(2)}_{1:l'+1}}$

11:             $w_{(1)} \leftarrow p(\overline{\zeta}^{(1)}_{l'+1}|\boldsymbol{x}), w_{(2)} \leftarrow p(\overline{\zeta}^{(2)}_{l'+1}|\boldsymbol{x})$

12:             $\boldsymbol{\mu}_* \leftarrow \frac{w_{(1)}\boldsymbol{\mu}_{(1)} + w_{(2)}\boldsymbol{\mu}_{(2)}}{w_{(1)} + w_{(2)}}, \boldsymbol{\sigma}^2_* \leftarrow \frac{w_{(1)}\boldsymbol{\sigma}^2_{(1)} + w_{(2)}\boldsymbol{\sigma}^2_{(2)}}{w_{(1)} + w_{(2)}}$

13:             $j_0 \leftarrow$ Maximum index for the child node whose parent path is $\overline{\zeta}^*_{1:l'}$

14:             $\overline{\zeta}^*_{1:l'+1} \leftarrow \left[\overline{\zeta}^*_{1:l'}, j_0 + 1\right]$

15:             $\text{N}(\overline{\zeta}^*_{1:l'+1}) \leftarrow \mathcal{N}(\boldsymbol{\mu}_*, \text{diag}(\boldsymbol{\sigma}^2_*))$

16:             $\text{N}(\overline{\zeta}^{(1)}_{1:l'+1}) \leftarrow \phi, \text{N}(\overline{\zeta}^{(2)}_{1:l'+1}) \leftarrow \phi$

17:             **if** $l' < L - 1$ **then**

18:                 $\mathbb{P}_{\text{inner}} \leftarrow \mathbb{P}_{\text{inner}} \cup \{\overline{\zeta}^*_{1:l'+1}\} \backslash \{\overline{\zeta}^{(1)}_{1:l'+1}, \overline{\zeta}^{(2)}_{1:l'+1}\}$

19:             **else**

20:                 $\mathbb{P}_{\text{full}} \leftarrow \mathbb{P}_{\text{full}} \cup \{\overline{\zeta}^*_{1:L}\} \backslash \{\overline{\zeta}^{(1)}_{1:L}, \overline{\zeta}^{(2)}_{1:L}\}$

21:             **end if**

22:             $Q' \leftarrow Q' \cup \{\overline{\zeta}^*_{1:l'+1}\}$

23:         **end for**

24:     **end if**

25:     *enqueue* $Q'$ to $\mathbb{Q}$

26:     **while** $Q_{\text{max}} < |\mathbb{Q}|$ **do** *dequeue* $\mathbb{Q}$

27:     $\mathbb{P} \leftarrow \mathbb{P}_{\text{full}} \cup \mathbb{P}_{\text{inner}}$

28:     $T \leftarrow (\mathbb{N}, \mathbb{P})$

29:     **return** $T, \mathbb{Q}$

30: **end function**

---

# D  NOTATIONS

The following Table 4 lists the notations used throughout this paper.

Table 4: Table of symbols

| Models | Symbol | Definition |
|---|---|---|
| All | $\boldsymbol{x}/\boldsymbol{x}'$ | An observed / reconstructed datapoint |
| | $\boldsymbol{z}$ | A latent representation |
| | $D/J$ | The input / latent dimensionality |
| | $g_*(\boldsymbol{x})$ | A encoder network parametrized by $*$, whose input is $\boldsymbol{x}$ |
| | $f_{\boldsymbol{\theta}}(\boldsymbol{z})$ | A decoder network parametrized by $\boldsymbol{\theta}$, whose input is $\boldsymbol{z}$ |
| | $\boldsymbol{\theta}$ | The variational parameters and weights of the decoder network $f_{\boldsymbol{\theta}}$ |
| | $\tilde{\boldsymbol{\mu}}_{\boldsymbol{z}}, \tilde{\boldsymbol{\sigma}}_{\boldsymbol{z}}^2$ | The variational mean and variance for Gaussian distribution $q_{\boldsymbol{\phi}_{\boldsymbol{z}}}(\boldsymbol{z}|\boldsymbol{x})$ |
| | $\boldsymbol{\mu}_{\boldsymbol{x}}, \boldsymbol{\sigma}_{\boldsymbol{x}}^2$ | The prior parameters, mean and variance, for Gaussian distribution $p_{\boldsymbol{\theta}}(\boldsymbol{x}|\boldsymbol{z})$ |
| VaDE & VAE-nCRP | $\boldsymbol{\phi}$ | The variational parameters and weights of the encoder network $g_{\boldsymbol{\phi}}$ |
| | $\tilde{\boldsymbol{\mu}}, \tilde{\boldsymbol{\sigma}}^2$ | The variational mean and variance for Gaussian distribution $q_{\boldsymbol{\phi}}(\boldsymbol{z}|\boldsymbol{x})$ |
| VaDE & HCRL | $N$ | The number of datapoints |
| | $\boldsymbol{x}_{n=1,\cdots,N}$ | $n$-th observed datapoint |
| | $\boldsymbol{z}_{n=1,\cdots,N}$ | $n$-th latent representation corresponding to $\boldsymbol{x}_n$ |
| VAE-nCRP & HCRL | $L$ | The height of the tree-based hierarchy |
| VaDE | $K$ | The number of (finite) clusters |
| | $c_{n=1,\cdots,N}$ | The cluster assignment of $\boldsymbol{z}_n, \in \{1,...,K\}$ |
| | $\boldsymbol{\kappa}$ | The prior parameter for multinomial distribution $p(\boldsymbol{c})$ |
| | $\boldsymbol{\mu}_c, \boldsymbol{\sigma}_c^2$ | The prior parameters, mean and variance, for Gaussian distribution of $c$-th cluster, $p(\boldsymbol{z})$ |
| VAE-nCRP | $M$ | The number of sequences |
| | $N_{m=1,\cdots,M}$ | The number datapoints in $m$-th sequence |
| | $\boldsymbol{x}_{m,n=1,\cdots,N}$ | $n$-th observed datapoint in $m$-th sequence |
| | $\boldsymbol{z}_{m,n=1,\cdots,N}$ | $n$-th latent representation corresponding to $\boldsymbol{x}_{mn}$ |
| | $v_{mp}$ | The Beta draws of $m$-th sequence on node $p$, for the tree-based stick-breaking construction |
| | $\gamma^*$ | The prior parameter for Beta distribution $p(v_{mp})$ |
| | $\gamma_{mp}^{(0)}, \gamma_{mp}^{(1)}$ | The variational parameters, for Beta distribution $q(v_{mp}|\boldsymbol{x}_m)$ |
| | $\boldsymbol{\zeta}_{mn}$ | The path assignment of $\boldsymbol{z}_{mn}$ |
| | $\boldsymbol{S}_{mn}^*$ | The variational parameter for multinomial distribution $q(\boldsymbol{\zeta}_{mn}|\boldsymbol{x}_{mn})$ |
| | $\boldsymbol{\alpha}_{\mathrm{par}(p)}$ | The $J$-dimensional parameter vector for the parent node of $p$ |
| | $\boldsymbol{\alpha}^*$ | The prior parameter for Gaussian distribution $p(\boldsymbol{\alpha}_p)$ for the root node |
| | $\boldsymbol{\mu}_{\mathrm{par}(p)}, \sigma_{\mathrm{par}(p)}^2$ | The variational mean and variance for Gaussian distribution $q(\boldsymbol{\alpha}_{\mathrm{par}(p)}|\boldsymbol{x})$ |
| | $\boldsymbol{\alpha}_p$ | The $J$-dimensional parameter vector for node $p$ |
| | $\sigma_N^2$ | The prior parameter, variance, for Gaussian distribution $p(\boldsymbol{\alpha}_p|\boldsymbol{\alpha}_{\mathrm{par}(p)})$ |
| | $\boldsymbol{\mu}_p, \sigma_p^2$ | The variational mean and variance for Gaussian distribution $q(\boldsymbol{\alpha}_p|\boldsymbol{x})$ |
| | $\sigma_D^2$ | The prior parameter, variance, for Gaussian distribution $p(\boldsymbol{z}_{mn}|\boldsymbol{\zeta}_{mn}, \boldsymbol{\alpha}_p)$ |
| HCRL | $\boldsymbol{\phi}_{\boldsymbol{z}}$ | The variational parameters and weights of the encoder network $g_{\boldsymbol{\phi}_{\boldsymbol{z}}}$ |
| | $\boldsymbol{\phi}_{\boldsymbol{\eta}}$ | The variational parameters and weights of the encoder network $g_{\boldsymbol{\phi}_{\boldsymbol{\eta}}}$ |
| | $\tilde{\boldsymbol{\mu}}_{\boldsymbol{z}}, \tilde{\boldsymbol{\sigma}}_{\boldsymbol{z}}^2$ | The variational mean and variance for Gaussian distribution $q_{\boldsymbol{\phi}_{\boldsymbol{z}}}(\boldsymbol{z}|\boldsymbol{x})$ |
| | $\tilde{\boldsymbol{\mu}}_{\boldsymbol{\eta}}, \tilde{\boldsymbol{\sigma}}_{\boldsymbol{\eta}}^2$ | The variational mean and variance for logistic normal distribution $q_{\boldsymbol{\phi}_{\boldsymbol{\eta}}}(\boldsymbol{\eta}|\boldsymbol{x})$ |
| | $\widetilde{\boldsymbol{\alpha}}$ | The variational parameter for Dirichlet distribution $q_{\boldsymbol{\phi}_{\boldsymbol{\eta}}}(\boldsymbol{\eta}|\boldsymbol{x})$ |
| | $v_i$ | The Beta draws for the tree-based stick-breaking construction of node $i$ |
| | $\gamma$ | The prior parameter for Beta distribution $p(v_i)$ |
| | $a_i, b_i$ | The variational parameters, for Beta distribution $q(v_i|\boldsymbol{x})$ |
| | $\boldsymbol{\zeta}_n$ | The path assignment of $\boldsymbol{z}_n$ |
| | $\boldsymbol{S}_n$ | The variational parameter for multinomial distribution $q(\boldsymbol{\zeta}_n|\boldsymbol{x}_n)$ |
| | $\boldsymbol{\eta}_n$ | The level proportion of $\boldsymbol{z}_n$ |
| | $\boldsymbol{\alpha}$ | The prior parameter for Dirichlet distribution $p(\boldsymbol{\eta}_n)$ |
| | $l_n$ | The level assignment of $\boldsymbol{z}_n, \in \{1,...,L\}$ |
| | $\boldsymbol{\omega}_n$ | The variational parameter for multinomial distribution $q(l_n|\boldsymbol{x}_n)$ |
| | $\boldsymbol{\mu}_i, \boldsymbol{\sigma}_i^2$ | The prior parameters, mean and variance, for Gaussian distribution of node $i$, $p(\boldsymbol{z}_n|\boldsymbol{\zeta}_n, \boldsymbol{\eta}_n)$ |

## E    GENERATIVE AND INFERENCE MODEL FOR HCRL

HCRL assumes the generative process as described in Section 3.1. Section E.1 describes the joint probability distribution, and Section E.2 presents the corresponding variational distributions. We adopt the much notation-related conventions from Wang & Blei (2009), especially on paths.

### E.1    GENERATIVE MODEL

$$p_{\boldsymbol{\theta}}(\boldsymbol{v},\boldsymbol{\zeta},\boldsymbol{\eta},\boldsymbol{l},\boldsymbol{z},\boldsymbol{x}) = p(\boldsymbol{v}|\gamma)\prod_{n=1}^{N}p(\boldsymbol{\zeta}_n|\boldsymbol{v})p(\boldsymbol{\eta}_n|\boldsymbol{\alpha})p(l_n|\boldsymbol{\eta}_n)p(\boldsymbol{z}_n|\boldsymbol{\zeta}_n,l_n,\boldsymbol{\mu}_{1:\infty},\boldsymbol{\sigma}_{1:\infty}^2)p_{\boldsymbol{\theta}}(\boldsymbol{x}_n|\boldsymbol{z}_n)$$

$$= \prod_{j\notin\mathcal{M}_T}p(v_j|\gamma)\prod_{i\in\mathcal{M}_T}p(v_i|\gamma)\prod_{n=1}^{N}p(\boldsymbol{\zeta}_n|\boldsymbol{v})p(\boldsymbol{\eta}_n|\boldsymbol{\alpha})p(l_n|\boldsymbol{\eta}_n)p(\boldsymbol{z}_n|\boldsymbol{\zeta}_n,l_n)p_{\boldsymbol{\theta}}(\boldsymbol{x}_n|\boldsymbol{z}_n)$$

- $\mathcal{M}_T$ : Set of all nodes in truncated tree $T$
- For $j\notin\mathcal{M}_T$, $p(v_j|\gamma)=\text{Beta}(v_j|1,\gamma)$
- For $i\in\mathcal{M}_T$, $p(v_i|\gamma)=\text{Beta}(v_i|1,\gamma)$
- $p(\boldsymbol{\zeta}_n=[1,\zeta_2,...,\zeta_L]|\boldsymbol{v})$

$$p(\boldsymbol{\zeta}_n=[1,\zeta_2,...,\zeta_L]|\boldsymbol{v}) = \prod_{l=1}^{L}\pi_{1,\zeta_2,...,\zeta_l}$$

$$= \prod_{l=1}^{L}\pi_{1,\zeta_2,...,\zeta_{l-1}}v_{1,\zeta_2,...,\zeta_l}\prod_{j=1}^{\zeta_l-1}(1-v_{1,\zeta_2,...,j})$$

$$= \prod_{l=1}^{L}\prod_{l'=1}^{l}\left\{v_{1,\zeta_2,...,\zeta_{l'}}\left(\prod_{j=1}^{\zeta_{l'}-1}(1-v_{1,\zeta_2,...,j})\right)\right\}$$

  – $\pi_{1,\zeta_2,...,\zeta_l}=\prod_{l'=1}^{l}\{v_{1,\zeta_2,...,\zeta_{l'}}(\prod_{j=1}^{\zeta_{l'}-1}(1-v_{1,\zeta_2,...,j}))\}$
- $p(\boldsymbol{\eta}_n|\boldsymbol{\alpha})=\text{Dirichlet}(\boldsymbol{\eta}_n|\boldsymbol{\alpha})$
- $p(l_n|\boldsymbol{\eta}_n)=\text{Multinomial}(\boldsymbol{\eta}_n)$
- $p(\boldsymbol{z}_n|\boldsymbol{\zeta}_n=\boldsymbol{\zeta},l_n=l)=\mathcal{N}(\boldsymbol{z}_n|\boldsymbol{\mu}_{\zeta_l},\boldsymbol{\sigma}_{\zeta_l}^2\boldsymbol{I}_J)$
- $p_{\boldsymbol{\theta}}(\boldsymbol{x}_n|\boldsymbol{z}_n)$ : Probabilistic decoding of $\boldsymbol{x}_n$ parametrized by $\boldsymbol{\theta}$, whose input is $\boldsymbol{z}_n$
- Tree-based stick-breaking construction
  – We will denote all Beta draws as $\boldsymbol{v}$, each of which is an independent draw from $\text{Beta}(\boldsymbol{v}|1,\gamma)$ (except for root $v_1=1$)
  * $v_i\sim\text{Beta}(v_i|1,\gamma)$
  – The root nodes stick length: $\pi_1=v_1\equiv 1$
  – Stick length at second level: $\pi_{1i}=\pi_1 v_{1i}\prod_{j=1}^{i-1}(1-v_{1j})$, $\sum_{i=1}^{\infty}\pi_{1i}=\pi_1=1$
  – For the segment $\pi_{1k}$, the stick lengths of its children are $\pi_{1ki}=\pi_{1k}v_{1ki}\prod_{j=1}^{i-1}(1-v_{1kj})$, for $i=1,2,...,\infty$, $\sum_{i=1}^{\infty}\pi_{1ki}=\pi_{1k}$

### E.2    INFERENCE MODEL

As VAE, we infer the random variables via the mean-field approximation, where the variational distribution, $q_{\boldsymbol{\phi}_\eta,\boldsymbol{\phi}_z}(\boldsymbol{v},\boldsymbol{\zeta},\boldsymbol{\eta},\boldsymbol{l},\boldsymbol{z}|\boldsymbol{x})$, approximates the intractable posterior. We model the variational distributions as follows:

$$q_{\boldsymbol{\phi}_\eta,\boldsymbol{\phi}_z}(\boldsymbol{v},\boldsymbol{\zeta},\boldsymbol{\eta},\boldsymbol{l},\boldsymbol{z}|\boldsymbol{x}) = q(\boldsymbol{v}|\boldsymbol{a},\boldsymbol{b},\boldsymbol{x})\prod_{n=1}^{N}q(\boldsymbol{\zeta}_n|\boldsymbol{x}_n)q_{\boldsymbol{\phi}_\eta}(\boldsymbol{\eta}_n|\boldsymbol{x}_n)q(l_n|\boldsymbol{\omega}_n,\boldsymbol{x}_n)q_{\boldsymbol{\phi}_z}(\boldsymbol{z}_n|\boldsymbol{x}_n)$$

$$= \prod_{j\notin\mathcal{M}_T}p(v_j|\gamma)\prod_{i\in\mathcal{M}_T}q(v_i|a_i,b_i)\prod_{n=1}^{N}q(\boldsymbol{\zeta}_n|\boldsymbol{x}_n)q_{\boldsymbol{\phi}_\eta}(\boldsymbol{\eta}_n|\boldsymbol{x}_n)q(l_n|\boldsymbol{\omega}_n,\boldsymbol{x}_n)q_{\boldsymbol{\phi}_z}(\boldsymbol{z}_n|\boldsymbol{x}_n)$$

- For $j \notin \mathcal{M}_T, p(v_j|\gamma) = \text{Beta}(v_j|1,\gamma)$

- For $i \in \mathcal{M}_T, q(v_i|a_i, b_i) \propto v_i^{a_i-1}(1-v_i)^{b_i-1} = \text{Beta}(v_i|a_i, b_i)$

  - $a_i = 1 + (L - l_i + 1)\sum_{n=1}^{N}\sum_{\zeta_{l_0+1},...,\zeta_L} q(\boldsymbol{\zeta}_n = [1, \zeta_2, ..., \zeta_{l_0}, \zeta_{l_0+1}, ..., \zeta_L])$

  - $b_i = \gamma + (L - l_i + 1)\sum_{n=1}^{N}\sum_{j,\zeta_{l_0+1},...,\zeta_L:j>\zeta_{l_0}} q(\boldsymbol{\zeta}_n = [1, \zeta_2, ..., \zeta_{l_0-1}, j, \zeta_{l_0+1}, ..., \zeta_L])$

    * $l_i$ : The level of the mixture component $i$

- $q(\boldsymbol{\zeta}_n|\boldsymbol{x}_n) \propto S_{n\overline{\zeta}} \triangleq \sum_{\boldsymbol{\zeta}\in\text{child}(\overline{\zeta})} S_{n\boldsymbol{\zeta}}$

  - $\overline{\zeta}$: a path in the truncated tree $T$, either an *inner path* (a path ending at an internal node) or a *full path* (a path ending at a leaf node)

  - child($\overline{\zeta}$): the set of all full paths that are not in $T$ but include $\overline{\zeta}$ as a sub path

    * As a special case, if $\overline{\zeta}$ is a full path, child($\overline{\zeta}$) just contains itself

  - In the case of a full path,

$$S_{n\overline{\zeta}} = S_{n\boldsymbol{\zeta}}$$
$$= \exp\left\{\mathbb{E}_q\left[\sum_{l=1}^{L}\sum_{l'=1}^{l}\left(\log v_{1,\zeta_2,...,\zeta_{l'}} + \sum_{j=1}^{\zeta_{l'}-1}\log(1 - v_{1,\zeta_2,...,j})\right)\right] + Z_0\right\}$$
$$= \exp\left\{\mathbb{E}_q\left[\sum_{l=1}^{L}\left(\sum_{l'=1}^{l}\left(\log v_{1,\zeta_2,...,\zeta_{l'}} + \sum_{j=1}^{\zeta_{l'}-1}\log(1 - v_{1,\zeta_2,...,j})\right) + \log\mathcal{N}(\boldsymbol{z}_n|\boldsymbol{\mu}_{\zeta_{nl}}, \boldsymbol{\sigma}^2_{\zeta_{nl}}\boldsymbol{I}_J)\right)\right]\right\}$$
$$= \exp\left\{\sum_{l=1}^{L}(L - l + 1)\left(\mathbb{E}_q[\log v_{1,\zeta_2,...,\zeta_l}] + \sum_{j=1}^{\zeta_l-1}\mathbb{E}_q[\log(1 - v_{1,\zeta_2,...,j})] + \log\mathcal{N}(\boldsymbol{z}_n|\boldsymbol{\mu}_{\zeta_{nl}}, \boldsymbol{\sigma}^2_{\zeta_{nl}}\boldsymbol{I}_J)\right)\right\}$$

  - In the case of an inner path, $\overline{\zeta} \triangleq [1, \overline{\zeta}_2, ..., \overline{\zeta}_{l_0}] \subset \mathcal{M}_T$

    * child($\overline{\zeta}$) $\triangleq \{[\overline{\zeta}, \overline{\zeta}_{l_0+1}, ..., \overline{\zeta}_L] : \overline{\zeta}_{l_0+1} > j_0\}$

    * $j_0$ : maximum index for the child node whose parent path is $\overline{\zeta}$

$$S_{n\overline{\zeta}} = \sum_{\boldsymbol{\zeta}\in\text{child}(\overline{\zeta})} S_{n\boldsymbol{\zeta}}$$
$$= \sum_{\boldsymbol{\zeta}\in\text{child}(\overline{\zeta})} \exp\left\{\mathbb{E}_q\left[\sum_{l=1}^{L}\left(\sum_{l'=1}^{l}\left(\log v_{1,\zeta_2,...,\zeta_{l'}} + \sum_{j=1}^{\zeta_{l'}-1}\log(1 - v_{1,\zeta_2,...,j})\right) + \log\mathcal{N}(\boldsymbol{z}_n|\boldsymbol{\mu}_{\zeta_{nl}}, \boldsymbol{\sigma}^2_{\zeta_{nl}}\boldsymbol{I}_J)\right)\right]\right\}$$
$$= \frac{\exp\left\{(\mathbb{E}_q[\sum_{l=l_0+1}^{L}\log\mathcal{N}(\boldsymbol{z}_n|\boldsymbol{\mu}_{\zeta_{nl}}, \boldsymbol{\sigma}^2_{\zeta_{nl}}\boldsymbol{I}_J)] + \sum_{l=l_0+1}^{L}\log(L - l + 1) + (L - l_0)(\psi(1) - \psi(1 + \gamma)))\right\}}{(1 - \exp\{\psi(\gamma) - \psi(1 + \gamma)\})^{L-l_0}}$$
$$\exp\left\{\sum_{l=1}^{l_0}(L - l + 1)\left(\mathbb{E}_q[\log v_{1,\zeta_2,...,\zeta_l}] + \sum_{j=1}^{\zeta_l-1}\mathbb{E}_q[\log(1 - v_{1,\zeta_2,...,j})]\right) + \log\mathcal{N}(\boldsymbol{z}_n|\boldsymbol{\mu}_{\zeta_{nl}}, \boldsymbol{\sigma}^2_{\zeta_{nl}}\boldsymbol{I}_J)\right\}$$
$$\exp\left\{\sum_{j=1}^{j_0}\mathbb{E}_q\left[(L - l_0)\log(1 - v_{1,\zeta_2,...,\zeta_{l_0},j})\right]\right\}$$

  - $q_{\phi_{\eta}}(\boldsymbol{\eta}_n|\boldsymbol{x}_n) = \text{Dirichlet}(\boldsymbol{\eta}_n|\widetilde{\boldsymbol{\alpha}}_n)$

    * $\widetilde{\alpha}_{nl} = \frac{1}{\widetilde{\sigma}^2_{\eta_{nl}}}(1 - \frac{2}{L} + \frac{e^{-\widetilde{\mu}_{\eta_{nl}}}}{L^2}\sum_{l'=1}^{L}e^{-\widetilde{\mu}_{\eta_{nl'}}})$

  - $q(l_n|\boldsymbol{\omega}_n, \boldsymbol{x}_n) = \text{Multinomial}(l_n|\boldsymbol{\omega}_n)$

    * $\omega_{nl} \propto \exp\{\psi(\widetilde{\alpha}_{nl}) - \psi(\widetilde{\alpha}_{n0}) + \sum_{\boldsymbol{\zeta}} S_{n\boldsymbol{\zeta}}(\sum_{j=1}^{J} -\frac{1}{2}\log(2\pi\sigma^2_{\zeta_{nl},j}) - \frac{\widetilde{\sigma}^2_{z_{nj}}}{2\sigma^2_{\zeta_{nl},j}} - \frac{(\widetilde{\mu}_{z_{nj}} - \mu_{\zeta_{nl},j})^2}{2\sigma^2_{\zeta_{nl},j}})\}$

      · $\widetilde{\alpha}_{n0} = \sum_{i=1}^{L}\widetilde{\alpha}_{ni}$

∗ Derivation for $\omega_{nl}$

$$\mathcal{L}_{\omega_{nl}} = \sum_l \omega_{nl}(\psi(\widetilde{\alpha}_{nl}) - \psi(\widetilde{\alpha}_{n0})) - \sum_l \omega_{nl} \log \omega_{nl}$$

$$+ \sum_{\boldsymbol{\zeta}'} S_{n\boldsymbol{\zeta}'} \left\{ \sum_l \omega_{nl} \left( \sum_{j=1}^J -\frac{1}{2} \log(2\pi\sigma^2_{\zeta'_{nl},j}) - \frac{\widetilde{\sigma}^2_{z_{nj}}}{2\sigma^2_{\zeta'_{nl},j}} - \frac{(\widetilde{\mu}_{z_{nj}} - \mu_{\zeta'_{nl},j})^2}{2\sigma^2_{\zeta'_{nl},j}} \right) \right\}$$

$$= \omega_{nl}(\psi(\widetilde{\alpha}_{nl}) - \psi(\widetilde{\alpha}_{n0})) + \sum_{\boldsymbol{\zeta}'} S_{n\boldsymbol{\zeta}'}\omega_{nl} \left( \sum_{j=1}^J -\frac{1}{2} \log(2\pi\sigma^2_{\zeta'_{nl},j}) - \frac{\widetilde{\sigma}^2_{z_{nj}}}{2\sigma^2_{\zeta'_{nl},j}} - \frac{(\widetilde{\mu}_{z_{nj}} - \mu_{\zeta'_{nl},j})^2}{2\sigma^2_{\zeta'_{nl},j}} \right)$$

$$- \omega_{nl} \log \omega_{nl} + \lambda(\sum_l \omega_{nl} - 1)$$

$$\frac{\partial \mathcal{L}_{\omega_{nl}}}{\partial \omega_{nl}} = (\psi(\widetilde{\alpha}_{nl}) - \psi(\widetilde{\alpha}_{n0})) + \sum_{\boldsymbol{\zeta}'} S_{n\boldsymbol{\zeta}'} \left( \sum_{j=1}^J -\frac{1}{2} \log(2\pi\sigma^2_{\zeta'_{nl},j}) - \frac{\widetilde{\sigma}^2_{z_{nj}}}{2\sigma^2_{\zeta'_{nl},j}} - \frac{(\widetilde{\mu}_{z_{nj}} - \mu_{\zeta'_{nl},j})^2}{2\sigma^2_{\zeta'_{nl},j}} \right)$$

$$- (\log \omega_{nl} + 1) + \lambda$$

$$\omega_{nl} = \exp\left\{ \psi(\widetilde{\alpha}_{nl}) - \psi(\widetilde{\alpha}_{n0}) + \sum_{\boldsymbol{\zeta}'} S_{n\boldsymbol{\zeta}'} \left( \sum_{j=1}^J -\frac{1}{2} \log(2\pi\sigma^2_{\zeta'_{nl},j}) - \frac{\widetilde{\sigma}^2_{z_{nj}}}{2\sigma^2_{\zeta'_{nl},j}} - \frac{(\widetilde{\mu}_{z_{nj}} - \mu_{\zeta'_{nl},j})^2}{2\sigma^2_{\zeta'_{nl},j}} \right) - 1 + \lambda \right\}$$

$$\omega_{nl} \propto \exp\left\{ \psi(\widetilde{\alpha}_{nl}) - \psi(\widetilde{\alpha}_{n0}) + \sum_{\boldsymbol{\zeta}'} S_{n\boldsymbol{\zeta}'} \left( \sum_{j=1}^J -\frac{1}{2} \log(2\pi\sigma^2_{\zeta'_{nl},j}) - \frac{\widetilde{\sigma}^2_{z_{nj}}}{2\sigma^2_{\zeta'_{nl},j}} - \frac{(\widetilde{\mu}_{z_{nj}} - \mu_{\zeta'_{nl},j})^2}{2\sigma^2_{\zeta'_{nl},j}} \right) \right\}$$

– $q_{\boldsymbol{\phi_z}}(\boldsymbol{z}_n|\boldsymbol{x}_n) = \mathcal{N}(\boldsymbol{z}_n|\widetilde{\boldsymbol{\mu}}_{\boldsymbol{z}_n}, \widetilde{\boldsymbol{\sigma}}^2_{\boldsymbol{z}_n}\boldsymbol{I}_J)$

## F    EVIDENCE LOWER BOUND

In this section, we present the detailed derivation of the ELBO in Equation 6, which is the objective function for learning HCRL.

$$\log p(\boldsymbol{x}) \geq \mathcal{L}_{ELBO}(\boldsymbol{x}) = \mathbb{E}_q\left[ \log \frac{p(\boldsymbol{v}, \boldsymbol{\zeta}, \boldsymbol{\eta}, \boldsymbol{l}, \boldsymbol{z}, \boldsymbol{x})}{q(\boldsymbol{v}, \boldsymbol{\zeta}, \boldsymbol{\eta}, \boldsymbol{l}, \boldsymbol{z}|\boldsymbol{x})} \right]$$

$$= \mathbb{E}_q\left[ \log \frac{\prod_{i \in \mathbb{M}_T} p(v_i|\gamma) \prod_{n=1}^N p(\boldsymbol{\zeta}_n|\boldsymbol{v})p(\boldsymbol{\eta}_n|\boldsymbol{\alpha})p(l_n|\boldsymbol{\eta}_n)p(\boldsymbol{z}_n|\boldsymbol{\zeta}_n, l_n)p_{\boldsymbol{\theta}}(\boldsymbol{x}_n|\boldsymbol{z}_n)}{\prod_{i \in \mathbb{M}_T} q(v_i|a_i, b_i) \prod_{n=1}^N q(\boldsymbol{\zeta}_n|\boldsymbol{x}_n)q_{\boldsymbol{\phi_\eta}}(\boldsymbol{\eta}_n|\boldsymbol{x}_n)q(l_n|\boldsymbol{\omega}_n, \boldsymbol{x}_n)q_{\boldsymbol{\phi_z}}(\boldsymbol{z}_n|\boldsymbol{x}_n)} \right]$$

$$= \sum_{i \in \mathbb{M}_T} \mathbb{E}_q[\log p(v_i|\gamma)] + \sum_{n=1}^N \mathbb{E}_q[\log p(\boldsymbol{\zeta}_n|\boldsymbol{v}) + \log p(\boldsymbol{\eta}_n|\boldsymbol{\alpha}) + \log p(l_n|\boldsymbol{\eta}_n) + \log p(\boldsymbol{z}_n|\boldsymbol{\zeta}_n, l_n)$$

$$+ \log p_{\boldsymbol{\theta}}(\boldsymbol{x}_n|\boldsymbol{z}_n)] - \sum_{i \in \mathbb{M}_T} \mathbb{E}_q[\log q(v_i|a_i, b_i)] - \sum_{n=1}^N \mathbb{E}_q[\log q(\boldsymbol{\zeta}_n|\boldsymbol{x}_n) + \log q_{\boldsymbol{\phi_\eta}}(\boldsymbol{\eta}_n|\boldsymbol{x}_n)$$

$$+ \log q(l_n|\boldsymbol{\omega}_n, \boldsymbol{x}_n) + \log q_{\boldsymbol{\phi_z}}(\boldsymbol{z}_n|\boldsymbol{x}_n)] \tag{6}$$

### F.1    DETAILED DERIVATION FOR ELBO

The followings are additional notations used for the detailed derivation:

- $\psi$ : The digamma function

- $\widetilde{\alpha}_{n0} = \sum_{i=1}^L \widetilde{\alpha}_{ni}, \alpha_0 = \sum_{i=1}^L \alpha_i$

$$
\begin{aligned}
\mathbb{E}_q[\log p(v_i|\gamma)] &= \int_{v'}\int_{\mathbf{z}'}\int_{\boldsymbol{\eta}'}\sum_{l'}\sum_{\boldsymbol{\zeta}'}\prod_{j\notin\mathcal{M}_T}p(v_j=v')\prod_{i\in\mathcal{M}_T}q(v_i=v')\prod_{n=1}^{N}q(\boldsymbol{\zeta}_n=\boldsymbol{\zeta}')q(\boldsymbol{\eta}_n=\boldsymbol{\eta}') \\
&\quad q(l_n=l')q(\mathbf{z}_n=\mathbf{z}')\log p(v_i=v')d\boldsymbol{\eta}'d\mathbf{z}'dv' \\
&= \int_{v'}\prod_{j\notin\mathcal{M}_T}p(v_j=v')\prod_{i\in\mathcal{M}_T}q(v_i=v')\log p(v_i=v')dv' \\
&= \int_{v'}q(v_i=v')\log p(v_i=v')dv' \\
&= \int_{v'}\text{Beta}(v'|a_i,b_i)\cdot\log\text{Beta}(v'|1,\gamma)dv' \\
&= \log\Gamma(1+\gamma)-\log\Gamma(1)-\log\Gamma(\gamma)+(1-1)\psi(a_i)+(\gamma-1)\psi(b_i) \\
&\quad +(-1-\gamma+2)\psi(a_i+b_i) \\
&= \log\Gamma(1+\gamma)-\log\Gamma(\gamma)+(\gamma-1)\psi(b_i)+(1-\gamma)\psi(a_i+b_i) \\
&= \log\Gamma(1+\gamma)-\log\Gamma(\gamma)+(\gamma-1)(\psi(b_i)-\psi(a_i+b_i)) \\
&= \log(\gamma\Gamma(\gamma))-\log\Gamma(\gamma)+(\gamma-1)(\psi(b_i)-\psi(a_i+b_i)) \\
&= \log\gamma+\log\Gamma(\gamma))-\log\Gamma(\gamma)+(\gamma-1)(\psi(b_i)-\psi(a_i+b_i)) \\
&= \log\gamma+(\gamma-1)(\psi(b_i)-\psi(a_i+b_i))
\end{aligned}
$$

$$
\begin{aligned}
\mathbb{E}_q[\log p(\boldsymbol{\zeta}_n|\boldsymbol{v})] &= \frac{\exp\{\log\Gamma(L-l_0+1)+(L-l_0)(\psi(1)-\psi(1+\gamma))\}}{(1-\exp\{\psi(\gamma)-\psi(1+\gamma)\})^{L-l_0}} \\
&\quad \times\ \exp\left\{\sum_{l=1}^{l_0}(L-l+1)\left(\mathbb{E}_q[\log v_{1,\zeta_2,\ldots,\zeta_l}]+\sum_{j=1}^{\zeta_l-1}\mathbb{E}_q[\log(1-v_{1,\zeta_2,\ldots,j})]\right)\right\} \\
&\quad \times\ \exp\left\{\mathbb{E}_q\left[(L-l_0)\sum_{j=1}^{j_0}\log(1-v_{1,\zeta_2,\ldots,\zeta_{l_0},j})\right]\right\}
\end{aligned}
$$

$$
\begin{aligned}
\mathbb{E}_q[\log p(\boldsymbol{\eta}_n|\boldsymbol{\alpha})] &= \int_{v'}\int_{\mathbf{z}'}\int_{\boldsymbol{\eta}'}\sum_{l'}\sum_{\boldsymbol{\zeta}'}\prod_{j\notin\mathcal{M}_T}p(v_j=v')\prod_{i\in\mathcal{M}_T}q(v_i=v')q(\boldsymbol{\zeta}_n=\boldsymbol{\zeta}')q(\boldsymbol{\eta}_n=\boldsymbol{\eta}') \\
&\quad q(l_n=l')q(\mathbf{z}_n=\mathbf{z})\log p(\boldsymbol{\eta}_n=\boldsymbol{\eta}'|\boldsymbol{\alpha})d\boldsymbol{\eta}'d\mathbf{z}'dv' \\
&= \int_{\boldsymbol{\eta}'}q(\boldsymbol{\eta}_n=\boldsymbol{\eta}')\log p(\boldsymbol{\eta}_n=\boldsymbol{\eta}'|\boldsymbol{\alpha})d\boldsymbol{\eta}'=\int_{\boldsymbol{\eta}'}\text{Dir}(\boldsymbol{\eta}'|\widetilde{\boldsymbol{\alpha}}_n)\cdot\log\text{Dir}(\boldsymbol{\eta}'|\boldsymbol{\alpha})d\boldsymbol{\eta}' \\
&= \log\Gamma(\alpha_0)-\sum_{i=1}^{L}\log\Gamma(\alpha_i)+\sum_{i=1}^{L}(\alpha_i-1)(\psi(\widetilde{\alpha}_{ni})-\psi(\widetilde{\alpha}_{n0}))
\end{aligned}
$$

$$
\begin{aligned}
\mathbb{E}_q[\log p(l_n|\boldsymbol{\eta}_n)] &= \int_{v'}\int_{\mathbf{z}'}\int_{\boldsymbol{\eta}'}\sum_{l'}\sum_{\boldsymbol{\zeta}'}\prod_{j\notin\mathcal{M}_T}p(v_j=v')\prod_{i\in\mathcal{M}_T}q(v_i=v')q(\boldsymbol{\zeta}_n=\boldsymbol{\zeta})q(\boldsymbol{\eta}_n=\boldsymbol{\eta}') \\
&\quad q(l_n=l')q(\mathbf{z}_n=\mathbf{z}')\log p(l_n=l'|\boldsymbol{\eta}_n)d\boldsymbol{\eta}d\mathbf{z}'dv' \\
&= \int_{\boldsymbol{\eta}'}\sum_{l'}q(\boldsymbol{\eta}_n=\boldsymbol{\eta}')q(l_n=l)\log p(l_n=l|\boldsymbol{\eta}_n=\boldsymbol{\eta}')d\boldsymbol{\eta}' \\
&= \sum_{l'}q(l_n=l')\int_{\boldsymbol{\eta}'}q(\boldsymbol{\eta}_n=\boldsymbol{\eta}')\log\text{Mult}(l'|\boldsymbol{\eta}')d\boldsymbol{\eta}' \\
&= \sum_{l'}\omega_{nl'}\int_{\boldsymbol{\eta}'}\text{Dir}(\boldsymbol{\eta}'|\widetilde{\boldsymbol{\alpha}}_n)\log\eta'_{l'}d\boldsymbol{\eta}' \\
&= \sum_{l'}\omega_{nl'}(\psi(\widetilde{\alpha}_{nl'})-\psi(\widetilde{\alpha}_{n0}))
\end{aligned}
$$

$$
\begin{aligned}
\mathbb{E}_q[\log p(\boldsymbol{z}_n|\boldsymbol{\zeta}_n, l_n)] &= \int_{v'}\int_{\boldsymbol{z}'}\int_{\boldsymbol{\eta}'}\sum_{l'}\sum_{\boldsymbol{\zeta}'}\prod_{j\notin\mathcal{M}_T}p(v_j = v')\prod_{i\in\mathcal{M}_T}q(v_i = v')q(\boldsymbol{\zeta}_n = \boldsymbol{\zeta})q(\boldsymbol{\eta}_n = \boldsymbol{\eta}') \\
&\qquad q(l_n = l')q(\boldsymbol{z}_n = \boldsymbol{z})\log p(\boldsymbol{z}_n = \boldsymbol{z}|\boldsymbol{\zeta}_n = \boldsymbol{\zeta}, l_n = l')d\boldsymbol{\eta}'d\boldsymbol{z}'dv' \\
&= \int_{\boldsymbol{z}'}\sum_{l'}\sum_{\boldsymbol{\zeta}'}q(\boldsymbol{\zeta}_n = \boldsymbol{\zeta})q(l_n = l')q(\boldsymbol{z}_n = \boldsymbol{z})\log p(\boldsymbol{z}_n = \boldsymbol{z}|\boldsymbol{\zeta}_n = \boldsymbol{\zeta}, l_n = l')d\boldsymbol{z}' \\
&= \sum_{l'}\sum_{\boldsymbol{\zeta}'}q(\boldsymbol{\zeta}_n = \boldsymbol{\zeta})q(l_n = l')\int_{\boldsymbol{z}'}q(\boldsymbol{z}_n = \boldsymbol{z})\log p(\boldsymbol{z}_n = \boldsymbol{z}|\boldsymbol{\zeta}_n = \boldsymbol{\zeta}, l_n = l')d\boldsymbol{z}' \\
&= \sum_{l'}\sum_{\boldsymbol{\zeta}'}q(\boldsymbol{\zeta}_n = \boldsymbol{\zeta}')q(l_n = l')\int_{\boldsymbol{z}'}\mathcal{N}(\boldsymbol{z}|\widetilde{\boldsymbol{\mu}}_{\boldsymbol{z}'}, \widetilde{\boldsymbol{\sigma}}_{\boldsymbol{z}}^2, \boldsymbol{I}_J)\cdot\log\mathcal{N}(\boldsymbol{z}|\boldsymbol{\mu}_{\zeta'_{nl'}}, \boldsymbol{\sigma}_{\zeta'_{nl'}}^2, \boldsymbol{I}_J)d\boldsymbol{z}' \\
&= \sum_{\boldsymbol{\zeta}'}S_{n\boldsymbol{\zeta}'}\left\{\sum_{l'}\omega_{nl'}\left(\sum_{j=1}^{J}-\frac{1}{2}\log(2\pi\sigma_{\zeta'_{nl'},j}^2) - \frac{\widetilde{\sigma}_{z_{nj}}^2}{2\sigma_{\zeta'_{nl'},j}^2} - \frac{(\widetilde{\mu}_{z_{nj}} - \mu_{\zeta'_{nl'},j})^2}{2\sigma_{\zeta'_{nl'},j}^2}\right)\right\}
\end{aligned}
$$

$$
\begin{aligned}
\mathbb{E}_q[\log p_{\boldsymbol{\theta}}(\boldsymbol{x}_n|\boldsymbol{z}_n)] &= \int_{v'}\int_{\boldsymbol{z}'}\int_{\boldsymbol{\eta}'}\sum_{l'}\sum_{\boldsymbol{\zeta}'}\prod_{j\notin\mathcal{M}_T}p(v_j = v')\prod_{i\in\mathcal{M}_T}q(v_i = v')q(\boldsymbol{\zeta}_n = \boldsymbol{\zeta})q(\boldsymbol{\eta}_n = \boldsymbol{\eta}') \\
&\qquad q(l_n = l')q(\boldsymbol{z}_n = \boldsymbol{z})\log p_{\boldsymbol{\theta}}(\boldsymbol{x}_n|\boldsymbol{z}_n = \boldsymbol{z}')d\boldsymbol{\eta}'d\boldsymbol{z}'dv' \\
&= \int_{\boldsymbol{z}'}q(\boldsymbol{z}_n = \boldsymbol{z})\log p_{\boldsymbol{\theta}}(\boldsymbol{x}_n|\boldsymbol{z}_n = \boldsymbol{z}')d\boldsymbol{z}' \\
&\approx \frac{1}{S}\sum_{s=1}^{S}\log p_{\boldsymbol{\theta}}(\boldsymbol{x}_n|\boldsymbol{z}_n^{(s)}), \text{ where } \boldsymbol{z}^{(i,s)} = \boldsymbol{\mu}_{\boldsymbol{x}}^{(i)} + \boldsymbol{\sigma}_{\boldsymbol{x}}^{(i)}\odot\boldsymbol{\epsilon}^{(s)} \text{ and } \boldsymbol{\epsilon}^{(s)}\sim\mathcal{N}(\boldsymbol{0}, \boldsymbol{I}_J) \\
&= \begin{cases}\frac{1}{S}\sum_{s=1}^{S}\sum_{d=1}^{D}x_{nd}\log\mu_{x_{nd}}^{(s)} + (1 - x_{nd})\log(1 - \mu_{x_{nd}}^{(s)}) & \text{if } \boldsymbol{x}_n \text{ is binary} \\ \frac{1}{S}\sum_{s=1}^{S}\sum_{d=1}^{D}-\frac{1}{2}\log(2\pi\sigma_{x_{nd}}^{(s)^2}) - \frac{(x_{nd} - \mu_{x_{nd}}^{(s)})^2}{2\sigma_{x_{nd}}^{(s)^2}} & \text{if } \boldsymbol{x}_n \text{ is real-valued}\end{cases}
\end{aligned}
$$

$$
\begin{aligned}
\mathbb{E}_q[\log q(v_i|a_i, b_i)] &= \int_{v'}\int_{\boldsymbol{z}'}\int_{\boldsymbol{\eta}'}\sum_{l'}\sum_{\boldsymbol{\zeta}'}\prod_{j\notin\mathcal{M}_T}p(v_j = v')\prod_{i\in\mathcal{M}_T}q(v_i = v')q(\boldsymbol{\zeta}_n = \boldsymbol{\zeta})q(\boldsymbol{\eta}_n = \boldsymbol{\eta}') \\
&\qquad q(l_n = l')q(\boldsymbol{z}_n = \boldsymbol{z})\log q(v_i = v')d\boldsymbol{\eta}'d\boldsymbol{z}'dv' \\
&= \int_{v'}\prod_{j\notin\mathcal{M}_T}p(v_j = v')\prod_{i\in\mathcal{M}_T}q(v_i = v')\log q(v_i = v')dv' \\
&= \int_{v'}q(v_i = v')\log q(v_i = v')dv' = \int_{v'}\text{Beta}(v'|a_i, b_i)\cdot\log\text{Beta}(v'|a_i, b_i)dv' \\
&= \log\Gamma(a_i + b_i) - \log\Gamma(a_i) - \log\Gamma(b_i) + (a_i - 1)\psi(a_i) + (b_i - 1)\psi(b_i) \\
&\quad + (-a_i - b_i + 2)\psi(a_i + b_i)
\end{aligned}
$$

$$
\begin{aligned}
\mathbb{E}_q[\log q(\boldsymbol{\zeta}_n|\boldsymbol{x}_n)] &= \int_{v'}\int_{\boldsymbol{z}'}\int_{\boldsymbol{\eta}'}\sum_{l'}\sum_{\boldsymbol{\zeta}}\prod_{j\notin\mathcal{M}_T}p(v_j = v')\prod_{i\in\mathcal{M}_T}q(v_i = v')q(\boldsymbol{\zeta}_n = \boldsymbol{\zeta})q(\boldsymbol{\eta}_n = \boldsymbol{\eta}') \\
&\qquad q(l_n = l')q(\boldsymbol{z}_n = \boldsymbol{z})\log q(\boldsymbol{\zeta}_n = \boldsymbol{\zeta})d\boldsymbol{\eta}'d\boldsymbol{z}'dv' \\
&= \sum_{\boldsymbol{\zeta}'}q(\boldsymbol{\zeta}_n = \boldsymbol{\zeta})\log q(\boldsymbol{\zeta}_n = \boldsymbol{\zeta}) \\
&= \sum_{\boldsymbol{\zeta}'}\frac{S_{n\boldsymbol{\zeta}'}}{\sum_{\boldsymbol{\zeta}''}S_{n\boldsymbol{\zeta}''}}\log\frac{S_{n\boldsymbol{\zeta}'}}{\sum_{\boldsymbol{\zeta}''}S_{n\boldsymbol{\zeta}''}}
\end{aligned}
$$

$$
\begin{aligned}
\mathbb{E}_q[\log q_{\boldsymbol{\phi}_{\boldsymbol{\eta}}}(\boldsymbol{\eta}_n|\boldsymbol{x}_n)] &= \int_{v'}\int_{\boldsymbol{z}'}\int_{\boldsymbol{\eta}'}\sum_{l'}\sum_{\boldsymbol{\zeta}'}\prod_{j\notin\mathcal{M}_T}p(v_j=v')\prod_{i\in\mathcal{M}_T}q(v_i=v')q(\boldsymbol{\zeta}_n=\boldsymbol{\zeta})q(\boldsymbol{\eta}_n=\boldsymbol{\eta}') \\
&\qquad q(l_n=l')q(\boldsymbol{z}_n=\boldsymbol{z})\log q(\boldsymbol{\eta}_n=\boldsymbol{\eta}')d\boldsymbol{\eta}'d\boldsymbol{z}'dv' \\
&= \int_{\boldsymbol{\eta}'}q(\boldsymbol{\eta}_n=\boldsymbol{\eta}')\log q(\boldsymbol{\eta}_n=\boldsymbol{\eta}')d\boldsymbol{\eta}' = \int_{\boldsymbol{\eta}'}\mathrm{Dir}(\boldsymbol{\eta}'|\widetilde{\boldsymbol{\alpha}}_n)\cdot\log\mathrm{Dir}(\boldsymbol{\eta}'|\widetilde{\boldsymbol{\alpha}}_n)d\boldsymbol{\eta}' \\
&= \log\Gamma(\widetilde{\alpha}_{n0}) - \sum_{i=1}^{L}\log\Gamma(\widetilde{\alpha}_{ni}) + \sum_{i=1}^{L}(\widetilde{\alpha}_{ni}-1)(\psi(\widetilde{\alpha}_{ni})-\psi(\widetilde{\alpha}_{n0}))
\end{aligned}
$$

$$
\begin{aligned}
\mathbb{E}_q[\log q(l_n|\boldsymbol{\omega}_n,\boldsymbol{x}_n)] &= \int_{v'}\int_{\boldsymbol{z}'}\int_{\boldsymbol{\eta}'}\sum_{l'}\sum_{\boldsymbol{\zeta}'}\prod_{j\notin\mathcal{M}_T}p(v_j=v')\prod_{i\in\mathcal{M}_T}q(v_i=v')q(\boldsymbol{\zeta}_n=\boldsymbol{\zeta})q(\boldsymbol{\eta}_n=\boldsymbol{\eta}') \\
&\qquad q(l_n=l')q(\boldsymbol{z}_n=\boldsymbol{z})\log q(l_n=l')d\boldsymbol{\eta}'d\boldsymbol{z}'dv' \\
&= \sum_{l'}q(l_n=l')\log q(l_n=l') = \sum_{l'}\mathrm{Mult}(l'|\boldsymbol{\omega}_n)\log\mathrm{Mult}(l'|\boldsymbol{\omega}_n) \\
&= \sum_{l'}\omega_{nl'}\cdot\log\omega_{nl'}
\end{aligned}
$$

$$
\begin{aligned}
\mathbb{E}_q[\log q_{\boldsymbol{\phi}_{\boldsymbol{z}}}(\boldsymbol{z}_n|\boldsymbol{x}_n)] &= \int_{v'}\int_{\boldsymbol{z}'}\int_{\boldsymbol{\eta}'}\sum_{l'}\sum_{\boldsymbol{\zeta}'}\prod_{j\notin\mathcal{M}_T}p(v_j=v')\prod_{i\in\mathcal{M}_T}q(v_i=v')q(\boldsymbol{\zeta}_n=\boldsymbol{\zeta})q(\boldsymbol{\eta}_n=\boldsymbol{\eta}') \\
&\qquad q(l_n=l')q(\boldsymbol{z}_n=\boldsymbol{z})\log q(\boldsymbol{z}_n=\boldsymbol{z})d\boldsymbol{\eta}'d\boldsymbol{z}'dv' \\
&= \int_{\boldsymbol{z}'}\int_{\boldsymbol{\eta}'}q(\boldsymbol{z}_n=\boldsymbol{z})\log q(\boldsymbol{z}_n=\boldsymbol{z})d\boldsymbol{\eta}'d\boldsymbol{z}' \\
&= \int_{\boldsymbol{z}'}q(\boldsymbol{z}_n=\boldsymbol{z})\log q(\boldsymbol{z}_n=\boldsymbol{z})d\boldsymbol{z}' = \int_{\boldsymbol{z}'}\mathcal{N}(\boldsymbol{z}|\widetilde{\boldsymbol{\mu}}_{\boldsymbol{z}'},\widetilde{\boldsymbol{\sigma}}_{\boldsymbol{z}'}^2,\boldsymbol{I}_J)\cdot\log\mathcal{N}(\boldsymbol{z}|\widetilde{\boldsymbol{\mu}}_{\boldsymbol{z}'},\widetilde{\boldsymbol{\sigma}}_{\boldsymbol{z}'}^2,\boldsymbol{I}_J)d\boldsymbol{z}' \\
&= -\frac{J}{2}\log(2\pi) - \frac{1}{2}\sum_{j=1}^{J}(1+\log\widetilde{\sigma}_{z_{nj}}^2)
\end{aligned}
$$

