# OpenReview forum: "Hierarchically Clustered Representation Learning"
_ICLR.cc/2019/Conference_

### Official Review · AnonReviewer1 · 2018-11-02
**Relevant problem, sound solution and convincing experimental evaluation. But limited novelty. Incremental work.**

**Rating:** 6
**Confidence:** 3

**Review:**

The paper presents a novel hierarchical clustering method over an embedding space. In the presented approach, both the embedding space and the hierarchical clustering are simultaneously learnt. The hierarchical clustering algorithm aims to recover complex clustering hierarchies which cannot be captured by previously proposed methods.

The paper address a relevant problem, which is of great interest for extracting knowledge from data. In general, the quality of the paper is high. The presented approach is based on a sound formalization of hierarchical clustering and deep generative models. The paper is easy to follow in spite of the technical difficulty. The experimental evaluation is really extensive. It compares against many state-of-the-art methods. And the results are promising from both a quantitative and qualitative point view.

The only issue with this paper is its degree of novelty, which is narrow. The proposed method adapt a previously presented hierarchical clustering method in the "standard space" (Griffiths et al., 2004) to an embedding space defined by a variational autoencoder model. The inference algorithm builds on standard techniques of deep generative models and, also, on previously proposed methods (Wand and Blei, 2003) for dealing with the complex hierarchical priors involved in this kind of models.

---

> ### Author Response · Authors · 2018-11-13
> **Response to Reviewer 1**
>
> [Q] The paper presents a novel hierarchical clustering method over an embedding space. In the presented approach, both the embedding space and the hierarchical clustering are simultaneously learned. The hierarchical clustering algorithm aims to recover complex clustering hierarchies which cannot be captured by previously proposed methods.
> [A1] Dear Reviewer 1, thank you for the thoughtful review. The reviewer mentioned our key point correctly. Many works on flat-clustered representation learning except for VAE-nCRP, has been limited to capture flat-level data structure.
>
> [Q] The paper address a relevant problem, which is of great interest for extracting knowledge from data.
> [A1] There are a lot of high-dimensional data around us, and it obviously contains complex structures inside. What we would like to argue through this study is that we can analyze the complex structure of data in the embedding space learned by a deep neural network.
>
> [Q] In general, the quality of the paper is high. The presented approach is based on a sound formalization of hierarchical clustering and deep generative models. The paper is easy to follow in spite of the technical difficulty. The experimental evaluation is really extensive. It compares against many state-of-the-art methods. And the results are promising from both a quantitative and qualitative point view.
> [A1] Thank you for the comment. As we assume a rather complex prior to embedding for flexibility, the technical depth of formalization has deepened. We concerned that it would be confused for the readers including the reviewers, to understand. Therefore, we carefully presented the figures, especially in Figure 3(a) showing an example of the variable values. In the case of experiments, we have devised various quantitative and qualitative experiments to assert why we need this hierarchically clustered representation learning. We empirically observed the performance improvement of both density estimation and hierarchical clustering, which motivates the joint optimization. Additionally, we qualitatively showed embedding plot, image generation, and result hierarchy with various datasets.
>
> [Q] The only issue with this paper is its degree of novelty, which is narrow. The proposed method adapt a previously presented hierarchical clustering method in the "standard space" (Griffiths et al., 2004) to an embedding space defined by a variational autoencoder model. The inference algorithm builds on standard techniques of deep generative models and, also, on previously proposed methods (Wand and Blei, 2003) for dealing with the complex hierarchical priors involved in this kind of models.
> [A1] As the reviewer pointed out, we adopted the partial components with the previously proposed techniques or methodologies. The theoretical contribution of our study can be considered in conjunction with the unified model based on the fully Bayesian approach of the probabilistic graphical model and the neural network. Additionally, we tuned the several detailed heuristic algorithms for operations such as GROW, PRUNE, and MERGE. Also, if we take a naive pipelined approach of iterative training between the hierarchical Gaussian mixture model and representation learning, then this work would be an obviously incremental work.
> [A2] VAE imposes a single Gaussian prior on embeddings, which leads to 1) the over-regularization, and 2) poor representations [1,2,5].
> [1] Chen, Xi, et al. "Infogan: Interpretable representation learning by information maximizing generative adversarial nets." NIPS. 2016.
> [2] Hoffman, Matthew D., and Matthew J. Johnson. "Elbo surgery: yet another way to carve up the variational evidence lower bound." Workshop in Advances in Approximate Bayesian Inference, NIPS. 2016.
> Therefore, the recently published researches can be divided into two branches: 1) designing of an objective function by introducing the additional regularized terms, or 2) constructing of a more flexible prior. Our work attempts to the latter approach, which proposes a new prior called a hierarchical-versioned Gaussian mixture distribution prior to the first trial of hierarchical density estimation in the embedding space. Another work of the latter approach is:
> - Variational Deep Embedding (VaDE) [3]: VAE+GMM
> - VAE-nCRP [4]: VAE+(nCRP+GMM)
> - VAE with a VampPrior [5]: VAE+ a variational mixture of posteriors prior.
> The contribution of these studies lies on 1) the formalization as a unified model based on the newly proposed prior, though not the original technique proposed by the authors, and 2) demonstrating the superiority of the prior.
> [3] Jiang, Zhuxi, et al. "Variational deep embedding: an unsupervised and generative approach to clustering." IJCAI, 2017.
> [4] Goyal, Prasoon, et al. "Nonparametric Variational Auto-Encoders for Hierarchical Representation Learning." ICCV. 2017.
> [5] Tomczak, Jakub, and Max Welling. "VAE with a VampPrior." AISTATS. 2018.
>
> Best regards,

---

### Official Review · AnonReviewer2 · 2018-11-05
**nested CRP plus neural network**

**Rating:** 5
**Confidence:** 4

**Review:**

The paper proposes using the nested CRP as a clustering model rather than a topic model. The clustering is on the latent vector input into a neural network for generating the observation. A variational approach is derived.

The proposed model seems like a straightforward extension of the nCRP with a deep model hanging off the end of it. A significant concern/confusion for me is that this doesn't seem to be a mixed membership model, and so I don't know how meaningful it is to generate a level distribution from a Dirichlet and then draw from that mixture one time. From the generative model it seems every data point has its own Dirichlet vector on levels. For topic models this makes sense since that vector is then drawn from multiple times (once per word) from a Discrete, so there's a distribution to actually learn. My understanding is that this isn't being done here.

---

> ### Author Response · Authors · 2018-11-13
> **Response to Reviewer 2**
>
> [Q] The paper proposes using the nested CRP as a clustering model rather than a topic model. The clustering is on the latent vector input into a neural network for generating the observation. A variational approach is derived. The proposed model seems like a straightforward extension of the nCRP with a deep model hanging off the end of it.
> [A1] Dear Reviewer 2, thank you for the thoughtful review. As the reviewer mentioned, we exploited the nested CRP prior to the path selection process. For performing a hierarchical density estimation task in embedding space, we additionally designed a hierarchical-versioned Gaussian mixture model prior with the nested CRP prior.
>
> [Q] A significant concern/confusion for me is that this doesn't seem to be a mixed membership model, and so I don't know how meaningful it is to generate a level distribution from a Dirichlet and then draw from that mixture one time. From the generative model, it seems every data point has its own Dirichlet vector on levels. For topic models, this makes sense since that vector is then drawn from multiple times (once per word) from a Discrete, so there's a distribution to actually learn. My understanding is that this isn't being done here.
> [A1] Thank you for the very constructive comments. In fact, we intended to model the level proportion as shown in the third part of our generative process on page 4. Often, for grouped-data, the level proportion (or topic proportion) is modeled as a group-specific variable. Under our non-grouped data setting, for example, two following approaches are possible: 1) as the reviewer mentioned, globally define a level proportion once, take multiple level samplings for each data, and 2) as our modeling, locally define the data-specific level proportion, followed by sampling the level (this is actually auxiliary variable for specifying the Gaussian distribution). The reason we chose the latter approach is for modeling more flexible prior. The Gaussian mixture distributions exist separately for each level, and we assume the generative process that the mixing coefficient for the level would be different for each data. Please consider that the data-instance we handled is a high-dimensional data of a document/an image rather than a word/a pixel. The hierarchically Gaussian mixture distributions are learned for different levels, and here assuming a common level proportion for all data forcefully limits the expressive power of the model. Also, for preventing the overfitting, we placed the common prior, Dirichlet(\alpha), on the data-specific level proportion, which can be considered as one of the regularization terms.
> [A2] Also, I would like to explain the reviewer’s comment as the formulae. The prior we suggested is this: \sum_{\zeta, l} nCRP(\zeta_n) * \eta_{nl} * Normal(-) ( please refer to the Figure 3(a).). Moreover, the point that the reviewer pointed out is on \eta_{nl}, i.e., ‘the reason for designing \eta as \eta_{nl}, why \eta is data-specific variable?’. There are similar works, which previously published [1-3]. They designed data-specific mixing coefficients of Gaussian mixture models, for improving more flexibility like ours.
> [1] Ban, Zhihua, Jianguo Liu, and Li Cao. "Superpixel Segmentation Using Gaussian Mixture Model." IEEE Transactions on Image Processing 27.8 (2018): 4105-4117.
> [2] Zhang, Hui, et al. "Automatic Visual Detection System of Railway Surface Defects With Curvature Filter and Improved Gaussian Mixture Model." IEEE Transactions on Instrumentation and Measurement 67.7 (2018): 1593-1608.
> [3] Ji, Zexuan, et al. "A spatially constrained generative asymmetric Gaussian mixture model for image segmentation." Journal of Visual Communication and Image Representation 40 (2016): 611-626.
> Under the newly proposed Gaussian mixture models from the above papers, the cluster assignment of data is sampled once from the data-specific mixing coefficient, where there is no theoretical problem as a fully Bayesian formalization.
> [A3] We were very impressed with the mathematical detail of the reviewer’s comment and thanked you to the reviewer again. If the reviewer agrees with our argument, we will reflect the argument in our paper.
>
> Best regards,

---

### Official Review · AnonReviewer3 · 2018-11-06
**Interesting and probably useful, but presentation needs work and there are some technical issues**

**Rating:** 5
**Confidence:** 4

**Review:**

This paper proposes using a variant of the nested CRP as a prior on the latent space of a variational autoencoder. The authors demonstrate that this approach is able to simultaneously learn a meaningful latent representation of high-dimensional data (text and images) and do hierarchical clustering in that space.

Pros:
* The high-level idea is compelling.
* The empirical results are compelling, and the evaluation is thorough.

Cons:
* The prose is pretty rough. The paper is full of sentences like "VAE-nCRP trade-off is the direct dependency modeling among clusters against the mean-field variational approach" that don't convey their intended meaning (at least to me).
* The random variable η seems completely superfluous. It only affects the likelihood through the level indicator l, but the marginal distribution p(l) = ∫_η p(η, l)dη \propto α is tractable, since only one level is drawn per observation. (This is not the case for the traditional nCRP as used in topic modeling, since there a different level is chosen for each word.)
* The novelty over the nCRP-VAE approach of Goyal et al. (2017) is pretty minor. The main difference seems to be that the model can select clusters at different levels, but I didn't quite get the intuition for why this should be desirable. In topic modeling, higher-level clusters tend to contain less-specialized words, and each document is a mix of specialized and general topics. But in this model, only one level is used to explain an entire image or document, and the idea that an entire image or document is much "more specialized" than another doesn't seem very intuitive to me.

---

> ### Author Response · Authors · 2018-11-13
> **Response to Reviewer 3 (2)**
>
> [Q] The novelty over the nCRP-VAE approach of Goyal et al. (2017) is pretty minor. The main difference seems to be that the model can select clusters at different levels, but I didn't quite get the intuition for why this should be desirable.
> [A1] The reviewers addressed one of the main differentiated features between VAE-nCRP and our model. We already described the directly related to the issue at the end of the fifth paragraph of Introduction:
> [A2] "Hierarchical mixture density estimation (Vasconcelos & Lippman, 1999), where all internal and leaf components are directly modeled to generate data, is a flexible framework for hierarchical mixture modeling, such as hierarchical topic modeling (Mimno et al., 2007; Griffiths et al., 2004), with regard to the learning of the internal components."
> [A3] In particular, (Mimno et al., 2007) [1] also argued that the modeling assumption of direct generation of data from internal components is a flexible framework, which strongly supports the generative process of our study. Besides, VAE-nCRP has a limitation in performing density estimation for embeddings only at the leaf-level, which contrasts with our model having the ability of hierarchical mixture density estimation.
> [1] Mimno, David, Wei Li, and Andrew McCallum. "Mixtures of hierarchical topics with pachinko allocation." Proceedings of the 24th international conference on Machine learning. ACM, 2007.
> [A4] We would like to stress that our theoretical contribution lies in the first trial of performing the hierarchical mixture density estimation in the embedding space with the unified model under the variational autoencoder framework.
>
> [Q] In topic modeling, higher-level clusters tend to contain less-specialized words, and each document is a mix of specialized and general topics. However, in this model, only one level is used to explain an entire image or document,
> [A1] This supports why we model the level proportion as a data-specific variable. We sample a level proportion for inferring the specialization level of the data and then, sample a level from the level proportion for specifying the Gaussian distribution among the Gaussian distributions lying on the sampled path.
> [A2] Also, I would like to explain the reviewer’s comment as the formulae. The prior we suggested is this: \sum_{\zeta, l} nCRP(\zeta_n) * \eta_{nl} * Normal(-) (-> please refer to the Figure 3(a).). Moreover, the point that the reviewer pointed out is on \eta_{nl}, i.e., ‘the reason for designing \eta as \eta_{nl}, why \eta is data-specific variable?’. There are similar works, which previously published [2-4]. They designed data-specific mixing coefficients of Gaussian mixture models, for improving more flexibility like ours.
> [2] Ban, Zhihua, Jianguo Liu, and Li Cao. "Superpixel Segmentation Using Gaussian Mixture Model." IEEE Transactions on Image Processing 27.8 (2018): 4105-4117.
> [3] Zhang, Hui, et al. "Automatic Visual Detection System of Railway Surface Defects With Curvature Filter and Improved Gaussian Mixture Model." IEEE Transactions on Instrumentation and Measurement 67.7 (2018): 1593-1608.
> [4] Ji, Zexuan, et al. "A spatially constrained generative asymmetric Gaussian mixture model for image segmentation." Journal of Visual Communication and Image Representation 40 (2016): 611-626.
> Under the newly proposed Gaussian mixture models from the above papers, the cluster assignment of data is sampled once from the data-specific mixing coefficient, where there is no theoretical problem as a fully Bayesian formalization.
> [A3] We were very impressed with the mathematical detail of the reviewer’s comment and thanked you to the reviewer again. If the reviewer agrees with our argument, we will reflect the argument in our paper.
>
> [Q] and the idea that an entire image or document is much "more specialized" than another doesn't seem very intuitive to me.
> [A1] We would like to illustrate with an example. Please refer to Figure 1 (a) and (b) in Introduction. From the Figure 1(a), we learned embeddings of digits 7, 4, and 9, which are very close to each other and inferred the several Gaussian mixture components such as internal components of 33, and leaf components of 66, 89, 60, and 88, which generate the embeddings with the high probabilities. Even with the same digit 7, 7 with the upper left edge, which is a shape shared with the digits 4 and 9, is generated from the internal mixture component, whereas the more specialized 7 containing the unique shape of digit 7 tend to be generated from the leaf mixture components.
>
> Best regards,

---

> ### Author Response · Authors · 2018-11-13
> **Response to Reviewer 3 (1)**
>
> [Q] This paper proposes using a variant of the nested CRP as a prior on the latent space of a variational autoencoder. The authors demonstrate that this approach is able to simultaneously learn a meaningful latent representation of high-dimensional data (text and images) and do hierarchical clustering in that space.
> [A1] Dear Reviewer 3, thank you for the constructive review. As you summarized out, we proposed a joint optimization of representation learning and hierarchical clustering in the embedding space. For the hierarchical clustering, we placed a hierarchical-versioned Gaussian mixture model prior, which is mentioned by the reviewer as ‘a variant of the nested CRP prior.’
>
> [Q] Pros: * The high-level idea is compelling. * The empirical results are compelling, and the evaluation is thorough.
> [A1] Thank you for your positive feedbacks. To assert the need for hierarchically clustered representation learning, we performed various quantitative and qualitative experiments with datasets from multiple domains.
>
> [Q] Cons: * The prose is pretty rough. The paper is full of sentences like "VAE-nCRP trade-off is the direct dependency modeling among clusters against the mean-field variational approach" that don't convey their intended meaning (at least to me). *
> [A1] Thank you for the reviewer's thoughtful comment. Especially, in the case of the given example sentence, we agree with the sentence should be supplemented. We will add more explanation at the end of the sentence after the reviewer’s agreement, and the sentence to be modified is as follows:
> [A2] "VAE-nCRP trade-off is the direct dependency modeling among clusters against the mean-field variational approach, which means that the joint distribution, q(\alpha_p, \alpha_par(p)) is no longer factorized like q(\alpha_p)q(\alpha_par(p))."
>
> [Q] The random variable η seems completely superfluous. It only affects the likelihood through the level indicator l, but the marginal distribution p(l) = ∫_η p(η, l)dη \propto α is tractable since only one level is drawn per observation. (This is not the case for the traditional nCRP as used in topic modeling since there a different level is chosen for each word.) *
> [A1] The issue “the marginal distribution p(l) = ∫_η p(η, l)dη \propto α is tractable, since only one level is drawn per observation.” that the reviewer mentioned is related to the conjugacy relationship between the categorical distribution (= one trial of multinomial distribution) and the Dirichlet distribution. The Dirichlet distribution is a conjugate prior for both the categorical distribution and the multinomial distribution regardless of the number of trials so that the \eta variable can be marginalized out.
> [A2] In fact, we intended to model the level proportion as shown in the third part of our generative process on page 4. Often, for grouped-data, the level proportion (or topic proportion) is modeled as a group-specific variable. Under our non-grouped data setting, for example, two following approaches are possible: 1) as the reviewer mentioned, globally define a level proportion once, take multiple level samplings for each data, and 2) as our modeling, locally define the data-specific level proportion, followed by sampling the level (this is actually auxiliary variable for specifying the Gaussian distribution). The reason we chose the latter approach is for modeling more flexible prior. The Gaussian mixture distributions exist separately for each level, and we assume the generative process that the mixing coefficient for the level would be different for each data. Please consider that the data-instance we handled is a high-dimensional data of a document/an image rather than a word/a pixel. The hierarchically Gaussian mixture distributions are learned for different levels, and here assuming a common level proportion for all data forcefully limits the expressive power of the model. Also, for preventing the overfitting, we placed the common prior, Dirichlet(\alpha), on the data-specific level proportion, which can be considered as one of the regularization terms.

---

### Meta-Review · Area_Chair1 · 2018-12-17
**metareview for representation learning paper**

**Confidence:** 4
**Recommendation:** Reject

**Metareview:**

While this was a borderline paper, concerns about the novelty and significance of the presented work exist on the part of all reviewers, and no reviewer was willing to argue for acceptance. Many good points to the work exist, and a stronger case on these issues would greatly strengthen the paper overall. I look forward to a future submission.